# Strong influence of trees outside forest in regulating microclimate of intensively modified Afromontane landscapes

Iris J. Aalto[1], Eduardo E. Maeda[1,2], Janne Heiskanen[1,3], Eljas K. Aalto[4], Petri K. E. Pellikka[1]

[1]Department of Geosciences and Geography, University of Helsinki, P.O. Box 64, FI-00014, Helsinki, Finland
[2]Area of Ecology and Biodiversity, School of Biological Sciences, Faculty of Science, University of Hong Kong, Hong Kong, SAR
[3]Institute for Atmospheric and Earth System Research, Faculty of Science, University of Helsinki, Finland
[4]Department of Economics, Turku School of Economics, 20014 University of Turku, Finland

*Correspondence to*: Iris Aalto (iris.aalto@helsinki.fi)

**Abstract.** Climate change is expected to have detrimental consequences on fragile ecosystems, threatening biodiversity as well as food security of millions of people. Trees are likely to play a central role in mitigating these impacts. The microclimatic conditions below tree canopies usually differ substantially from the ambient macroclimate, as vegetation can buffer temperature changes and variability. Trees cool down their surroundings through several biophysical mechanisms, and the cooling benefits occur also with trees outside forest. The aim of this study was to examine the effect of canopy cover on microclimate in an intensively modified Afromontane landscape in Taita Taveta, Kenya. We studied temperatures recorded by 19 microclimate sensors under different canopy covers, and land surface temperature (LST) estimated by Landsat 8 thermal infrared sensor. We combined the temperature records with high–resolution airborne laser scanning data to untangle the combined effects of topography and canopy cover on microclimate. We developed four multivariate regression models to study the joint impacts of topography and canopy cover on LST. The results showed a negative linear relationship between canopy cover percentage and daytime mean ($R^2 = 0.65$) and maximum ($R^2 = 0.75$) temperatures. Any increase in canopy cover contributed to reducing temperatures. The average difference between 0 % and 100 % canopy cover sites was 5.2 ˚C in mean temperatures and 10.2 ˚C in maximum temperatures. Canopy cover reduced LST on average by 0.05 ˚C/%CC. The influence of canopy cover on microclimate was shown to vary strongly with elevation and ambient temperatures. These results demonstrate that trees have substantial effect on microclimate, but the effect is dependent on macroclimate, highlighting the importance of maintaining tree cover particularly in warmer conditions. Hence, we demonstrate that trees outside forests can increase climate change resilience in fragmented landscapes, having strong potential for regulating regional and local temperatures.

**Keywords**

Agroforestry, airborne laser scanning, canopy cover, land surface temperature, Landsat 8, microclimate

## 1. Introduction

Climate change poses an imminent threat to the rich biodiversity and frequently found fragile socio–economic conditions that characterize Afromontane ecosystems and their surroundings. In these regions, climate warming is mostly driven by land use and land cover change (LULCC) (IPCC, 2018; Pellikka and Hakala, 2019; Abera et al., 2020). Agricultural expansion, in particular, has caused rapid loss of tropical forests (FAO, 2016). Forests are essential in mitigating climate warming, due to their role in especially the carbon and water cycles (Beer et al., 2010; Ellison et al., 2017; De Frenne et al., 2019).

Currently, forests cover approximately 4 billion hectares of the Earth's surface (FAO, 2016). Forests are often defined as a land area of at least 0.5 hectares with a minimum canopy cover of 10 % and trees higher than 5 m (FAO, 2015). Trees that are not part of a forest are commonly called "trees outside forest" (TOF) and, by the definition of FAO (2000), include trees on farmland, in cities, and in other locations not defined as forest. Forests and TOF provide vital ecosystem services including water regulation, air purification, carbon sequestration, and climate regulation (Chakravarty et al., 2019; Kuyah et al., 2019; Skole et al., 2021). They are also a source of goods for humans, such as food and timber (Thijs et al., 2015; Martínez Pastur et al., 2018; Chakravarty et al., 2019). As global forest cover decreases, the importance of TOF will increase in biodiversity conservation and ecosystem service provision (Mace et al., 2012; Mendenhall et al., 2016), and TOF can be beneficial in reducing the pressure on native forests (Ilyama et al., 2014; Chakravarty et al., 2019). For example, in Taita Hills in Kenya, TOF make up a remarkable amount of the area's total aboveground carbon and play an important part in carbon sequestration in the area (Pellikka et al., 2018), especially because Taita Hills have experienced massive indigenous forest loss since 1950's (Pellikka et al., 2009). Forest loss is a major threat to biodiversity, as Taita Hills are identified as an important biodiversity hotspot (Pellikka et al., 2013; Thijs et al., 2015). Biodiversity is considered fundamental for the provision of ecosystem services (Mace et al., 2012).

Many ecosystem services, such as nutrient cycling and pollination, occur in the understories, where tree canopies create the appropriate microclimates essential for these processes (De Frenne et al., 2013). The term "microclimate" describes the climatic conditions near the ground or along the vertical forest profile, experienced by terrestrial organisms (De Frenne et al., 2019; Zellweger et al., 2019). In contrast to free air temperatures, which are highly controlled by elevation and atmospheric processes, temperatures close to the ground are primarily affected by topographic factors and vegetation structures that produce local microclimates through shading, mixing of air, and evapotranspiration (Geiger, 1980; Das et al., 2015; Zellweger et al., 2020). Climatic conditions below forest canopies can vary spatially within the forest (Chen et

al., 1999) and differ substantially from the ambient macroclimate: this difference is referred to as microclimatic buffering (Ewers and Banks-Leite, 2013; Zellweger et al., 2020). The temperature buffering provided by tree cover may protect ecosystems from climate change consequences (Zomer et al., 2016; Ellison et al., 2017; De Frenne et al., 2019; Wanderley et al., 2019), but the magnitude of the buffering is affected by the forest area (Ewers and Banks-Leite, 2013). In time, forest microclimates will likely warm like the macroclimate around them, and fragmentation may accelerate this process (Ewers and Banks-Leite, 2013; Li et al., 2016).

Despite wide recognition of the vital role microclimates play, studies about tropical forests' response to climate warming have primarily focused on the macroscale (Belsky et al., 1989; De Frenne et al., 2019, Wild et al., 2019). Weather stations that commonly measure free air temperatures at 1.5 meters height do not capture microclimatic conditions that are ecologically more relevant to terrestrial organisms (Potter et al., 2013; Wild et al., 2019; Maclean et al., 2021). Further, microclimate may be a better indicator of how well forests mitigate climate change than macroclimate (De Frenne et al., 2013). Due to the importance of microclimatic conditions for the survival of tropical species facing climate change, below–canopy microclimates warrant further investigation (Potter et al., 2013; Jucker et al., 2018; De Frenne et al., 2021). In our study area in Kenya, temperatures are expected to increase by 2–4 °C by the end of the century (Adhikari et al., 2015), and changes in precipitation, that will increase the moisture stress of crops, are projected (MoALF, 2016). Dry spells, heat stress and extreme rain events pose a threat to the area's agricultural production. These phenomena cause crop failure and low yields, and hence affect the livelihoods of people (Adhikari et al., 2015; MoALF 2016). Farmers have already noticed climate fluctuations that affect both crops and livestock in the area (Mwalusepo et al., 2015).

Microclimatic studies require extensive field measurements, making them sometimes unpractical or imprecise in larger scale applications (Prata et al., 1995). Alternatively, measuring satellite–derived land surface temperature (LST) proves useful when point-wise field measurements are insufficient, given the high spatial coverage of spaceborne LST and the strong correlation between LST and air temperature (Jin and Dickinson, 2010; Li et al., 2013). These two measurements differ in their physical principles: air temperature is the kinetic temperature of the air, whereas LST is defined as the radiometric temperature recorded by a satellite sensor in a scale of the sensor's pixel size (Jin and Dickinson, 2010). Various factors affect LST: atmospheric conditions, water content of the surface, topography and canopy cover control the energy exchange processes (Goward and Hope, 1989; Nemani et al., 1993), which makes accurate estimation of LST a challenge (Simó et al., 2018; Li et al., 2013). Vegetation density has a strong negative relationship with LST due to evapotranspiration causing increased latent heat loss from the canopy (Goward et al., 1985; Goward and Hope, 1989; Nemani and Running, 1997). Canopies' cooling effect has different magnitudes at different latitudes: for example, tropical forests experience the strongest cooling effect (Li et al., 2015; Wanderley et al., 2019).

In remote sensing of vegetation, common outputs in previous research are land cover and land use types or vegetation indices, such as the normalized vegetation index (NDVI) or leaf area index (LAI) (Nemani et al., 1993; Kim 2013; He et al., 2019). However, airborne laser scanning (ALS) has proven to be a more effective method for computing structural variables, such as above-ground biomass, canopy height, and canopy cover (Griffin et al., 2008; Heiskanen et al., 2015a; Heiskanen et al., 2015b; Pellikka et al., 2018; Jucker et al., 2018). Canopy cover (CC) describes the proportion of the forest floor covered by the vertical projection of the tree crowns (Korhonen et al., 2006) and it is the most important variable used in defining forests or other land with tree cover (FAO, 2015). ALS can assess tree cover over large areas more precisely than field measurements can. Hence, when ALS is combined with either field-based or remotely sensed temperatures, we can study the influence of trees on temperature in a new way of that is both nuanced and large scale. The complexity of the issue with climate change requires attention at both spatial resolutions.

The primary objective of this study was to examine how different levels of CC can contribute to lower temperatures and more stable microclimates across a highly heterogeneous Afromontane landscape in Kenya. We based our analysis on micro-climatological measurements and CC estimates retrieved from ALS data. Microclimate sensors cannot entirely capture the spatial variability of temperatures, especially in heterogeneous landscapes. Therefore, we used satellite thermal data to provide a comprehensive and spatially continuous representation of the relationship between CC and temperature.

## 2. Materials and methods

### 2.1 Study area

The Taita Hills are located in the Taita-Taveta County in the Coast Province in southern Kenya (3° 25′ S, 38° 20′ E), approximately 200 km from Mombasa and 360 km from the capital city Nairobi. The study area comprises of the Taita Hills and the lowland areas of Maktau, LUMO Community Wildlife Sanctuary and Taita Hills Wildlife Sanctuary that have been laser scanned by the University of Helsinki (Fig. 1). The elevation in the study area varies from 640 m in the lowlands to the highest peak of the hills, Vuria, at 2208 m. Climate is mainly semi-arid. According to the Kenya Ministry of Agriculture, Livestock and Fishery (MoALF), annual precipitation averages 650 mm, but differences between hills and lowlands are notable: lowlands receive 500 mm annually compared to 1500 mm in the hills. Two rainy seasons control the climate and growing seasons: long rains from March to June, and short rains from October to December (Pellikka et al., 2013), while months from January to March are a short hot dry season and months from June to October long cool dry season (Wachiye et al., 2020). Mean temperature in the lowlands is 23 °C and in the hills 18 °C (MoALF, 2016).

Vegetation varies from dry savanna and shrubland in the lowlands dominated by *Vachellia ssp*. and *Commiphora ssp*. tree species to indigenous cloud forests in the hilltops. Small indigenous forest fragments, exotic tree plantations, and intensive agriculture dominate the landscape in the hills. Agroforestry practices are typical, which increases cropland CC.

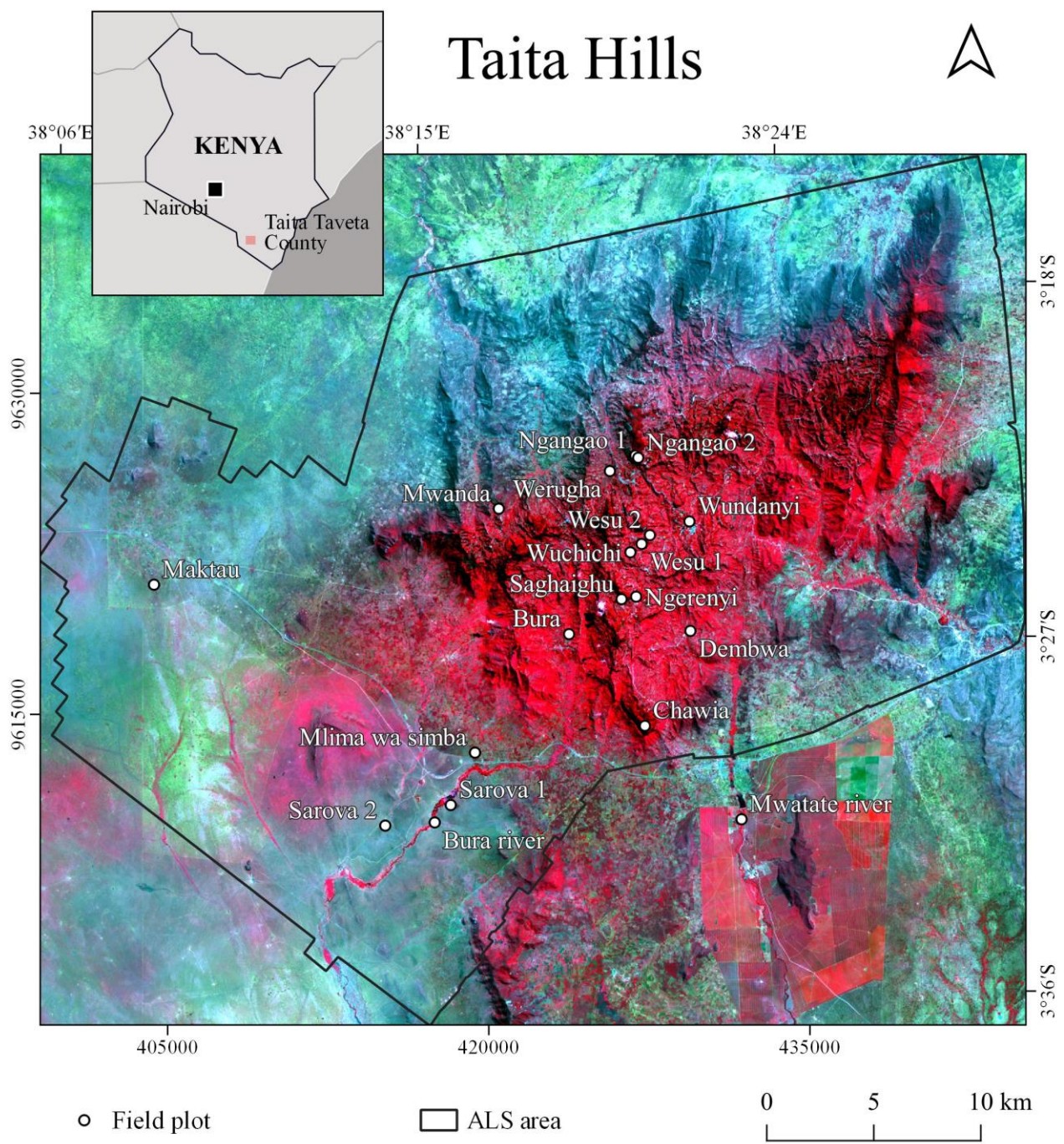

**Figure 1:** Field plots with microclimate sensors in Taita Taveta County, Kenya. ALS refers to airborne laser scanning. The base map is a false color Landsat 8 OLI image from July 4, 2019.

126

**2.2 Airborne laser scanning data**

We applied an ALS-based Digital Elevation Model (DEM) raster at 1 m resolution and a CC raster at 30 m resolution. The ALS data for the hills were acquired in February 2014 and February 2015, and the data for lowland areas in March 2014. The mean pulse density of the ALS data in the hills was 3.1 pulses/$m^{-2}$ and mean return density 3.4 returns/$m^{-2}$, for the lowlands the pulse density was 1.04 pulses/$m^{-2}$. The ALS data used in this study are described in detail in Adhikari et al. (2017) and Amara et al. (2020) with the description of pre-processing and derivation of DEM and CC rasters.

We resampled the DEM to 30 m resolution to fit to the spatial resolution of the Landsat 8 image, and utilized it to derive topographic factors slope degree (°) and aspect (°) using ArcGIS Pro spatial analyst tools.

**2.3 Microclimatological field measurements**

Based on the CC raster derived from the ALS data, we selected a total of 19 field plots representing different CC levels (Table 1). In the plots, we installed TOMST TMS-4 microclimate sensors to measure temperature at three different heights: soil at 6 cm below ground, surface at 2 cm above ground, and air temperature at 15 cm above ground ($T_{soil}$, $T_{surface}$ and $T_{air}$, respectively) (Wild et al., 2019). The sensors were deployed in places that were as flat as possible to reduce the effect of slope, and that received both sunlight and shade during the day with the changing sun angles. In high CC sites, the sensors were shaded most of the day, while in the open areas, the sensors were exposed to sunlight all day.

The sensors measured parameters every 15 minutes from June 13 to July 10, 2019. We calculated daytime temperature aggregates between sunrise and sunset, local time 06.30–18.30 UTC + 3h. We calculated maxima as the mean of daily maxima, and minimum temperatures as the mean of minimum temperatures based on the 24 hour cycle.

To isolate the influence of CC on microclimate, we quantified and later removed the effect of topography, such as elevation (m) and slope (°), on temperature. We examined the relationships between the variables first with Pearson's correlation using elevation, slope and CC as explanatory variables in a multiple regression model. Elevation and CC were the only statistically significant variables. We corrected the daytime mean temperatures according to the altitudinal lapse rates, which were 7.26 °C $km^{-1}$ for soil temperature ($T_{soil}$), 8.09 °C $km^{-1}$ for surface temperature ($T_{surface}$) and 8.06 °C $km^{-1}$ for air temperature ($T_{air}$). In the case of diurnal analysis, we applied separate lapse rates for each hour that were derived from the regression analyses. The lapse rates were 6.1 °C–8.2 °C $km^{-1}$ in $T_{soil}$, 3.8 °C–10.4 °C $km^{-1}$ in $T_{surface}$, and 3.3 °C–10.2 °C $km^{-1}$ in $T_{air}$. To find the relationships between temperature, CC and topographic variables, we conducted statistical analyses, including descriptive statistics, linear regression and Pearson's correlation. We used standard deviation (SD) to describe the variability of temperatures. We used RStudio (R Core Team, 2019) for all statistical analyses.

The ALS data was 4–5 years older than the field measurements. Moreover, the ALS data was collected during the short dry season, in contrast to the field measurements, which we carried out during the start of the long dry season in June 2019. To address the mismatch between the data collection dates, we acquired hemispherical photography at each field plot for validating the CC raster. The differences in CC were not statistically significant and we considered the estimates consistent enough for proceeding the analysis using CC from ALS. In the case of Mwatate river plot, CC was retrieved by hemispherical photography only, because the plot was outside of the ALS coverage. The methodology is described in Appendix A.

| Site | CC % | Elevation, m | Description |
| --- | --- | --- | --- |
| Bura | 68 | 1095 | Parkland by school campus |
| Bura river | 79 | 880 | Riverine forest |
| Chawia | 97 | 1562 | Indigenous forest |
| Dembwa | 13 | 1083 | Agroforestry |
| Maktau | 19 | 1044 | Bushland |
| Mlima wa simba | 8 | 923 | Bushland |
| Mwanda | 2 | 1653 | Bushland |
| Mwatate river | 63 | 884 | Riverine forest |
| Ngangao 1 | 94 | 1775 | Indigenous forest |
| Ngangao 2 | 77 | 1778 | Eucalyptus forest |
| Ngerenyi campus | 44 | 1572 | Macadamia plantation |
| Saghaighu | 16 | 1611 | Agroforestry |
| Sarova 1 | 0 | 901 | Bushland |
| Sarova 2 | 0 | 900 | Grassland |
| Werugha | 8 | 1613 | Macadamia plantation |
| Wesu 1 | 53 | 1642 | Forest edge |
| Wesu 2 | 0 | 1562 | Open maize field |
| Wuchichi | 36 | 1595 | Agroforestry |
| Wundanyi | 31 | 1372 | Riverside bushland |

**Table 1:** Names, canopy cover (CC) percentages, elevations and descriptions of field plot sites.

**2.4 Land surface temperature**

To observe the effect of CC on temperature in Taita Taveta County, we applied Landsat 8 OLI thermal infrared sensor (TIRS) satellite image data, downloaded from USGS Earth Explorer (https://earthexplorer.usgs.gov/). The bands 10 and 11 of TIRS provide thermal infrared imagery in a resolution of 100 m, but we resampled the band to 30 m to concert with the OLI images. The image used in the study was a Level-1 scene obtained on July 4, 2019 at approximately 10:30 UTC + 3h with solar azimuth angle of 45.6° and solar elevation angle of 52.1°. The cloud cover of the whole scene was 11.67 %; there was no completely cloudless scene over the study area for the timing of the field measurements.

Several methods have been developed to retrieve LST from Landsat 8. Unfortunately, shortly after the launch of Landsat 8 in 2013, a stray light problem was detected with TIRS band 11, and it was not recommended by United States Geological Survey (USGS) to apply for scientific purposes (USGS, 2017). We applied the workflow by Ndossi and Avdan (2016) and used the single channel (SC) method by Jiménez-Muñoz and Sobrino (2003) to calculate LST, because SC method needs only one thermal infrared channel, and land surface emissivity (LSE) and water vapor content as parameters. Using only one channel may introduce uncertainty in LST estimations: for Landsat 8 band 10, Jiménez-Muñoz et al. (2014) reported RMSE = 1.5 K, while in Ndossi and Avdan (2016) the RMSE = 3.06 °C. Nevertheless, SC method is most accurate for sensors with effective wavelengths near to 11 μm (Jiménez-Muñoz et al., 2014), the wavelength of Landsat 8 band 10 being 10.6–11.19 μm.

We calculated LSE using the algorithm based on the NDVI image, where pixels were given pre-defined emissivity values based on the NDVI derived from the red, green and infrared bands. Please refer to Ndossi and Avdan (2016) for more details. Water vapor content at the time of the satellite overpass was 1.7 g cm$^{-2}$, and was calculated with Eq. (1) using the relative humidity and temperature data obtained from the local weather station:

$$w = 0.0981 \times \left\{ 10 \times 0.6108 \times \exp\left[\frac{17.27 \times (T_0 - 273.15)}{237.3 + (T_0 - 273.15)}\right] \times RH \right\} + 0.1679 \tag{1}$$

where $w$ = water vapor content, $T_0$ = air temperature and $RH$ = relative humidity.

The SC formula is shown in Eq. (2):

$$T_s = \gamma \left[\frac{1}{\varepsilon}(\Psi_1 L_{sen} + \Psi_2) + \Psi_3\right] + \delta \tag{2}$$

$$\gamma = \frac{T_{sen}^2}{b_\gamma L_{sen}} \tag{3}$$

$$\delta = T_{sen} - \frac{T_{sen}^2}{b_\gamma} \tag{4}$$

where $Ts$ = LST, $\gamma$ = parameter depending on Eq. (3), $\delta$ = parameter depending on Eq. (4), $\varepsilon$ = land surface emissivity,
$L_{sen}$ = top of atmosphere spectral radiance (W sr$^{-1}$ m$^{-2}$ μm$^{-1}$), $b\gamma$ = 1324 K for Landsat 8 band 10, and $T_{sen}$ = at sensor
brightness temperature (K). We obtained the atmospheric parameters Ψ1, Ψ2 and Ψ3 with Eq. (5):

$$\begin{bmatrix} \Psi_1 \\ \Psi_2 \\ \Psi_3 \end{bmatrix} = \begin{bmatrix} c_{11} & c_{12} & c_{13} \\ c_{21} & c_{22} & c_{23} \\ c_{31} & c_{32} & c_{33} \end{bmatrix} \begin{bmatrix} \omega^2 \\ \omega \\ 1 \end{bmatrix} \tag{5}$$

According to Jiménez-Muñoz, et al. (2014), the coefficients for atmospheric parameters for Landsat 8 TIRS are as in Eq.

194 (6):

$$c = \begin{bmatrix} 0.04019 & 0.02916 & 1.01523 \\ -0.38333 & -1.50294 & 0.20324 \\ 0.00918 & 1.36072 & -0.27514 \end{bmatrix} \tag{6}$$

We conducted similar topographic correction with the Landsat image as with microclimate sensors to exclude the effect
of topography on LST. Topographic variables (elevation, slope and aspect), CC, and their interaction terms were included
as independent factors and LST as the dependent factor in four multiple regression models (Table 2). We classified aspect
to nine classes indicating eight cardinal directions (south, south-west, west, north-west, north, north-east, east, south-
east), and flat surface. The classes were treated as dummy variables due to their categorical nature. We also classified
elevation to three classes: below 1000 m, 1000–1500 m, and above 1500 m. We used the LST at elevation of 880 m, slope
of 0 ° and aspect class north as reference.

| Model | Predictors |
|---|---|
| 1 | DEM, CC, slope, aspect (south, south-west, west, north-west, north, north-east, east, south-east) |
| 2 | DEM, CC, slope, aspect (south, south-west, west, north-west, north, north-east, east, south-east) elevation zones (<1000 m, 1000–1500 m, >1500 m), elevation zones * CC |
| 3 | DEM, CC, slope, aspect (south, south-west, west, north-west, north, north-east, east, south-east), DEM * CC |
| 4 | DEM, CC, slope, aspect (south, south-west, west, north-west, north, north-east, east, south-east), slope * aspect classes, elevation zones (<1000 m, 1000–1500 m, >1500 m), elevation zones * CC |

**Table 2:** Topographic and canopy cover (CC) predictors included in the four multiple regression models used in the
analysis of Landsat 8 land surface temperature.

**3. Results**
**3.1 Canopy cover and microclimate**
**3.1.1 Mean, maximum and minimum temperatures**
Topographically corrected mean temperatures (T') had significant negative correlation with CC at all the measurement
heights (T'$_{surface}$ and T'$_{air}$ $r$ = -0.84, T'$_{soil}$ $r$ = -0.78). Based on the linear regression, an increase from 0 % to 100 % CC
decreased T'$_{soil}$ by 5.2 °C ($R^2$ = 0.6), T'$_{surface}$ by 5.9 °C ($R^2$ = 0.71) and T'$_{air}$ by 4.6 °C ($R^2$ = 0.71) (Fig. 2). The average
effect on combined T'$_{soil}$, T'$_{surface}$ and T'$_{air}$ was 5.2 °C ($R^2$ = 0.68). T'$_{surface}$ and T'$_{air}$ were in general higher than T'$_{soil}$.
CC also affected variability of mean temperatures: SD of temperatures decreased by approximately 0.1 per 10 CC%
increase at all measurement heights (Fig. 2). In T'$_{air}$, the relationship was not as evident as in T'$_{soil}$ and T'$_{surface}$: SD
decreased distinctly first when CC% was higher than 60 %.
CC had a strong effect on maximum temperatures at all measurement heights, T'$_{surface}$ being affected the most. High CC
sites experienced the lowest T'$_{surface}$ and T'$_{air}$ maxima, while T'$_{surface}$ and T'$_{air}$ were the hottest in Maktau and sites with 0
% CC. Here, topographically corrected average maximum temperatures ranged between 30 °C and 38.5 °C. Again, T'$_{surface}$
and T'$_{air}$ were generally higher than T'$_{soil}$. The linear models showed that the increase from 0 % CC to 100 % CC decreased
the maximum T'$_{soil}$ by 9 °C ($R^2$ = 0.69), T'$_{surface}$ by 12.1 °C ($R^2$ = 0.74) and T'$_{air}$ by 9.6 °C ($R^2$ = 0.69) (Fig. 3). On average,
the difference was 10.2 °C. Similarly to mean temperatures, SD of maximum temperatures decreased with increasing CC:
T'$_{soil}$ showed a more gradual decrease than T'$_{soil}$ and T'$_{surface}$, where SD decreased substantially only in high CC sites (Fig.
3). The SD of maximum temperatures were higher than in mean temperatures.
Based on the regression coefficients, which indicate the magnitude of the influence of CC on temperature, the cooling
effect of CC was stronger on maximum temperatures than mean. Additionally, whereas CC affected mean T'$_{soil}$ more than
mean T'$_{air}$, in maximum temperatures the situation was the opposite, and T'$_{air}$ was more affected by CC than T'$_{soil}$ (Fig. 2
and Fig. 3).

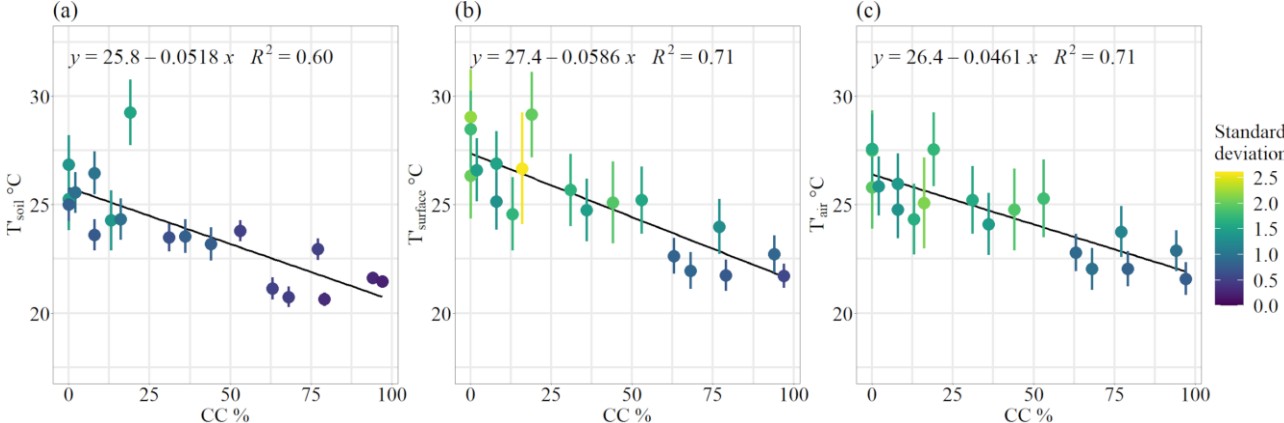


**Figure 2:** Scatterplots of topographically corrected daytime mean temperatures (T') and standard deviation against

canopy cover (CC) percentage, with regression line. a) Soil temperature. b) Surface temperature. c) Air temperature.

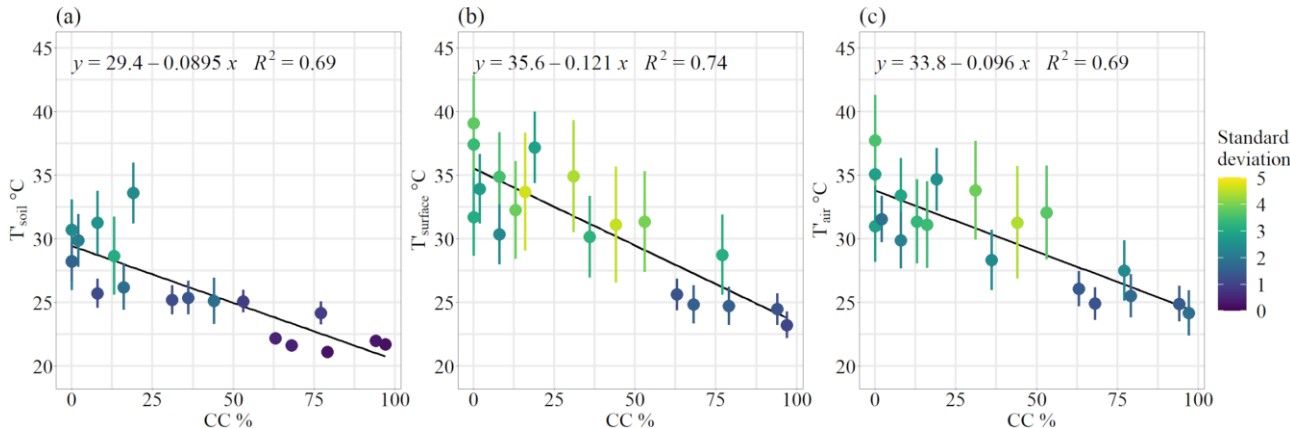


**Figure 3:** Scatterplots of topographically corrected daytime maximum temperatures (T') and standard deviation against

canopy cover (CC) percentage, with regression line. a) Soil temperature. b) Surface temperature. c) Air temperature.
Minimum temperatures showed no explicit relationship with CC, and sites with similar CC had high temperature
variability. $R^2$ were low ($< 0.2$) at all measurement heights, and correlations between temperatures and CC were
insignificant. All results from the regression analyses are summarized in Table 3.

| | Measurement height | Max (C°) | Site, CC % | Min (C°) | Site, CC % | Coef | $R^2$ | r | p-value |
|---|---|---|---|---|---|---|---|---|---|
| **Mean** | $T'_{soil}$ | 29.3 | Maktau, 19 % | 20.6 | Bura river, 79 % | -0.052 | 0.604 | -0.777 | <0.001* |
| | $T'_{surface}$ | 29.2 | Maktau, 19 % | 21.7 | Chawia, 97 % | -0.059 | 0.711 | -0.843 | <0.001* |
| | $T'_{air}$ | 27.6 | Sarova 2, 0 % | 21.6 | Chawia, 97 % | -0.046 | 0.710 | -0.842 | <0.001* |
| **Maximum** | $T'_{soil}$ | 33.3 | Maktau, 19 % | 20.8 | Bura river,79 % | -0.09 | 0.693 | -0.832 | <0.001* |
| | $T'_{surface}$ | 38.8 | Sarova 2, 0 % | 22.9 | Chawia ,97 % | -0.121 | 0.742 | -0.862 | <0.001* |
| | $T'_{air}$ | 37.4 | Sarova 2, 0 % | 23.8 | Chawia, 97 % | -0.1 | 0.686 | -0.828 | <0.001* |
| **Minimum** | $T'_{soil}$ | 23.0 | Maktau, 19 % | 19.2 | Bura, 68 % | -0.003 | 0.083 | -0.289 | 0.231 |
| | $T'_{surface}$ | 19.5 | Chawia, 97 % | 12.9 | Sarova 2, 0 % | -0.024 | 0.189 | 0.435 | 0.063 |
| | $T'_{air}$ | 19.3 | Ngangao 2, 77 % | 12.3 | Sarova 2, 0 % | -0.023 | 0.149 | 0.386 | 0.102 |

**Table 3.** Topographically corrected temperature (T') statistics for the soil, surface and air. Temperatures in the maximum and minimum columns refer to the highest and lowest mean, maximum and minimum temperatures. Site refers to where the highest and lowest temperatures were measured and their respective canopy cover (CC) percentage. * indicates statistical significance.

### 3.1.2 Temporal variation

Figure 4 presents the daily variation in topographically corrected daytime mean temperatures. The effect of CC was evident at all three measurement heights: mean temperatures were lower in high CC sites than in open areas, yet some low CC sites exhibited relatively low temperatures. For example, on July 2, which was one of the hottest days of the study period, temperature differences between the hottest (Maktau, 19 % CC) and coolest (Ngangao 1, 94 % CC) sites were 11.0 °C in $T'_{soil}$, 11.3 °C in $T'_{surface}$ and 9.8 °C in $T'_{air}$. Even during the coldest days, temperatures were lower in sites with dense canopies than in open land. Especially $T'_{soil}$ in the sites with high CC remained relatively stable from day to day, showing little fluctuation even during the hot day streaks: differences in mean temperatures remained even less than 1 °C between hottest and coolest days.

The cooling effect of CC varied throughout the study period: on hot days, the cooling effect (described by CC's regression coefficient in Fig. 4) increased, while on cooler days, the cooling effect decreased. The strongest cooling took place in

T'$_{surface}$ on June 23, when CC's cooling effect was 7.6 °C. T'$_{surface}$ had overall the highest cooling effect (3.3 °C–7.6 °C)
and T'$_{air}$ the weakest (2.6 °C–6 °C). In T'$_{soil}$, the cooling effect was 3.2 °C–6.9 °C (Fig. 4).

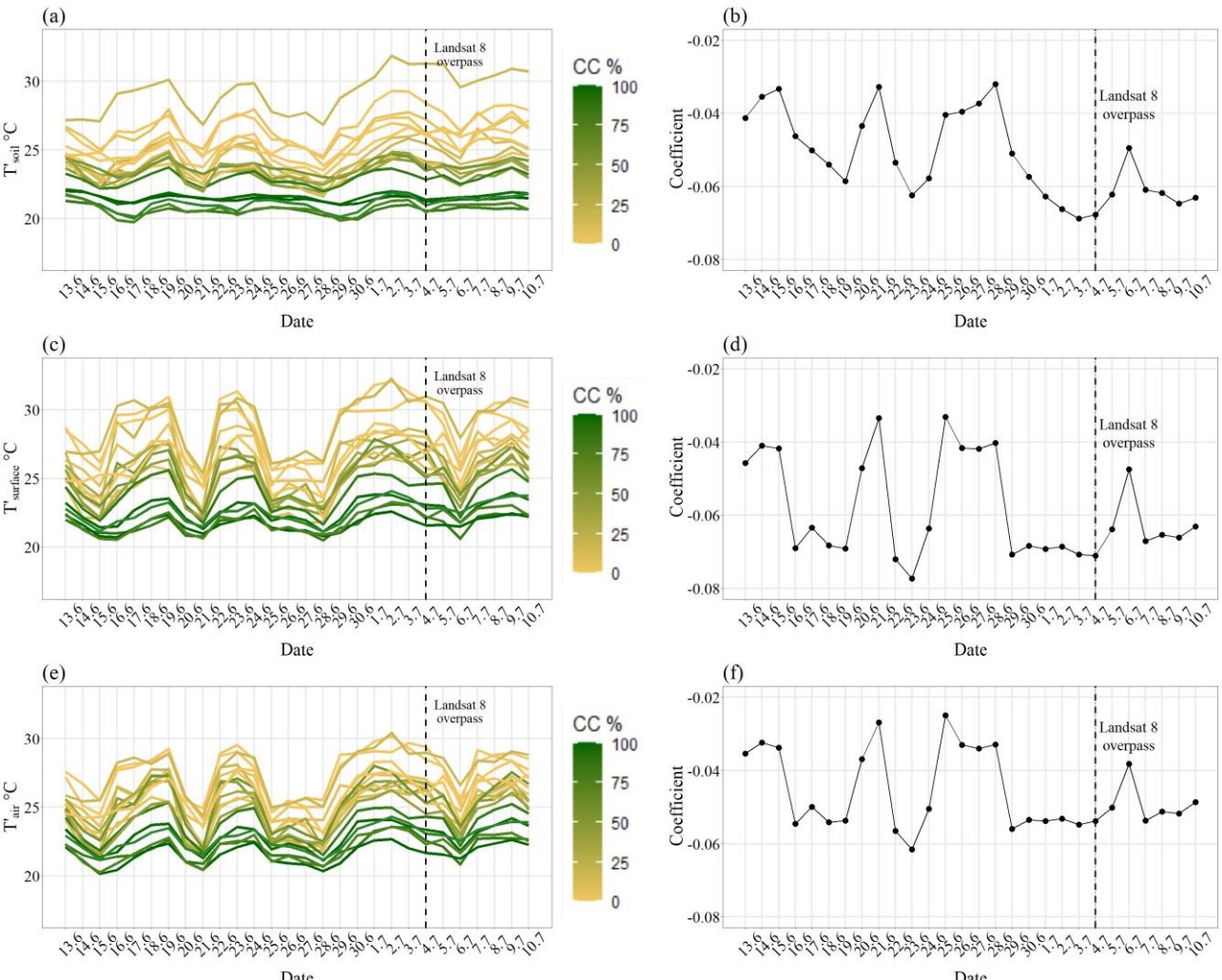


**Figure 4**: Daily variation in topographically corrected daytime (6.30–18.30) mean temperatures (T') between June 13
and July 10, 2019 (left), and cooling effect of canopy cover (described by regression coefficient) (right). Line color
indicates canopy cover (CC) percentage. Dashed line represents the overpass date of Landsat 8, July 4, 2019. a–b) Soil
temperature. c–d) Surface temperature. e–f) Air temperature.
Figure 5 shows the intra-daily temperature variability based on study period means. T'$_{soil}$ were more stable than T'$_{surface}$
and T'$_{air}$ that showed higher peaks and drops. In the morning, temperatures at all measurement heights started to rise
rapidly between 6:00 and 8:00. Changes in T'$_{soil}$ seemed to lag a couple of hours behind T'$_{surface}$ and T'$_{air}$: they reached
highest readings between 11:00 and 15:00, while T'$_{soil}$ peaked between 15:00 and 17:00. Further, after peaking,
temperatures decreased before stabilizing between 19:00 and 20:00 in T'$_{surface}$ and T'$_{air}$, while T'$_{soil}$ decreased slower. T$_{soil}$
remained warmer during the night than the other two measurement heights.

Figure 5 also describes the correlation between CC and temperatures. The impact of CC was the lowest in the morning, when the temperatures also reached their minima. The strongest correlation ($r < -0.8$) occurred during afternoon at all measurement heights. T'$_{soil}$ correlated negatively with CC throughout the day, in contrast to T'$_{surface}$ and T'$_{air}$, where correlations were positive during the night.

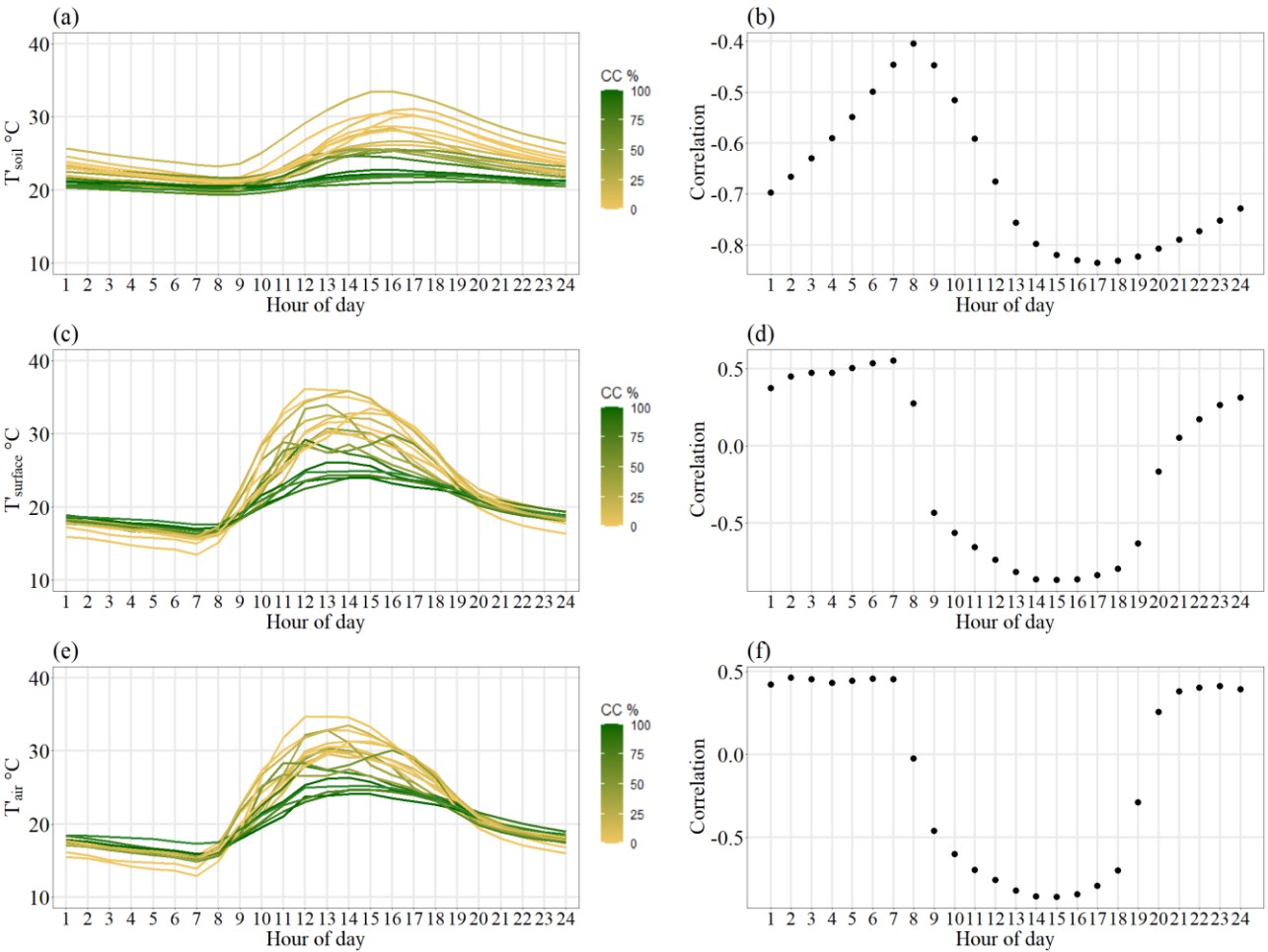

**Figure 5**: Topographically corrected diurnal mean temperatures (T') (left) and the correlation between T' and canopy cover (CC) percentage (right) between June 13 and July 10, 2019. Hour refers to ordinal number of hour, e.g. 1 means 00:00–01:00. Line color indicates CC percentage. a–b) Soil temperature. c–d) Surface temperature. e–f) Air temperature.

**3.2 Landsat 8 land surface temperature**

**3.2.1 Land surface temperature compared with temperatures measured in the field**

LST and raw field temperatures (T) at the time of satellite overpass showed statistically significant correlation ($r = 0.82$, 0.79 and 0.84 at T$_{soil}$, T$_{surface}$ and T$_{air}$, respectively) (Fig. 6). At 18 sites out of 19, LST was higher than T$_{soil}$, whereas

between LST and $T_{surface}$ or $T_{air}$ there was no consistent difference. Mean differences were 4.1 °C ($T_{soil}$), -0.03 °C ($T_{surface}$)
and 0.57 °C ($T_{air}$). The $T_{soil}$ difference was statistically significant with 95 % confidence, while $T_{surface}$ and $T_{air}$ not.

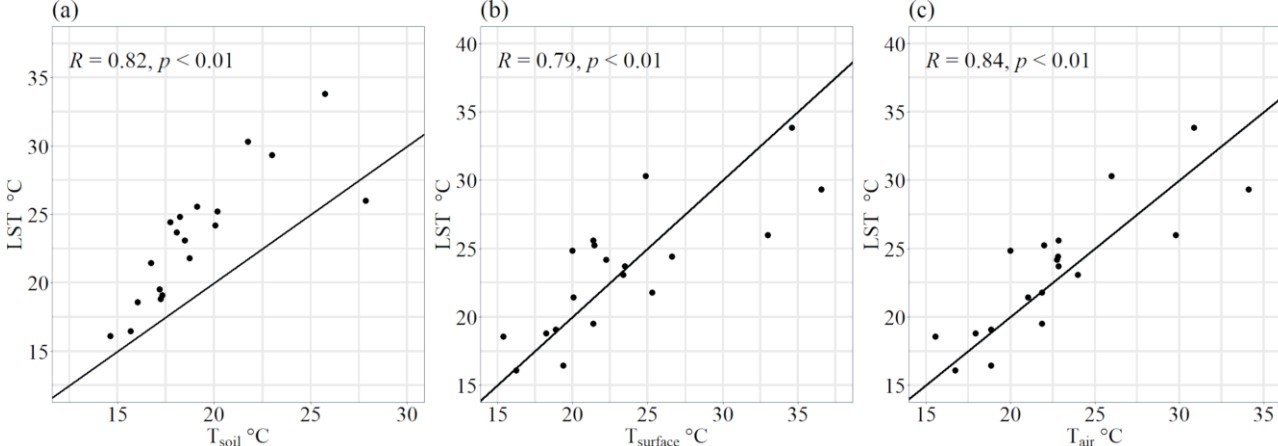


**Figure 6:** Landsat 8 land surface temperature (LST) compared with raw field temperatures (T) at the time of satellite
overpass (10:30) on July 4, 2019. a) LST and soil temperature. b) LST and surface temperature. c) LST and air
temperature.

**3.2.2 Impact of canopy cover and topography on land surface temperature**
Topographic variables elevation, slope and aspect had all a significant effect on LST. In all four models, the elevational
lapse rates varied from 11 °C km$^{-1}$ to 15 °C km$^{-1}$. Aspect, in turn, had a varying impact depending on the model, but the
general trend was that south, south-west and west had the highest cooling, as was expected at the time of the day. The
effect of slope decreased as the models became more complex, and the joint impacts of slope and aspect in Model 4 were
greater than the effects of slope or aspect alone. The results of all four models can be found in Appendix B.
All the variables in Model 1 showed statistical significance ($R^2 = 0.74$). Based on the regression analysis, generally the
increase from 0 % CC to 100 % CC decreased LST with 5 °C. After the exclusion of other variables except CC, correlation
between LST and CC was -0.37 (p < 0.001) and $R^2 = 0.14$.
In Model 2, three elevation zones (below 1000 m, 1000–1500 m, above 1500 m) were added to the model. This increased
the $R^2$ to 0.77, demonstrating a notable difference in the cooling effect of CC depending on elevation zone. At the
elevations below 1000 m, the cooling effect of CC when moving from  0 % CC to 100% CC was 6.8 °C, between 1000–
1500 m the effect was 3.7 °C, and above 1500 m the effect was 4 °C. Roughly, the cooling impact of CC above 1000 m
decreased to almost half of the impact in the lowlands.
In Model 3, the interaction term of CC and elevation zones was replaced with interaction term of CC and the continuous
variable elevation from the DEM. This produced $R^2 = 0.74$. The coefficient for the interaction term was 0.00005,
indicating that an increase of 1000 m in elevation decreased the cooling effect of CC by 0.05 °C. The model performed
poorer compared to Model 2.
Model 4 was built up on Model 2 by adding interaction terms between slope and aspect classes. Model 4 performed best
of the four ($R^2 = 0.77$), but the difference was not large compared to Model 2. The cooling effect of CC in the lowlands
was 6.8 °C, the same as in Model 2. In the elevation zone 1000–1500 m the cooling effect was 3.7 °C and above 1500 m
it was 3 °C. The cooling effect of CC in 1000–1500 m had the same magnitude as in Model 2, and it decreased by further
0.7 °C in elevations above 1500 m.
In summary, including either of the elevation factors (DEM or elevation zones) in the model showed that elevation
affected CC's cooling effect significantly, having almost two times higher impact in the lowlands compared to the hills.
The dependence of CC's impact on elevation is demonstrated in Fig. 7 using eight elevation classes. CC's regression
coefficients decreased with increasing elevation after 1000 m, yet increased again between 1200–1400 m to roughly the
same as in the lowlands. The effect was the smallest in elevations above 1800 m.

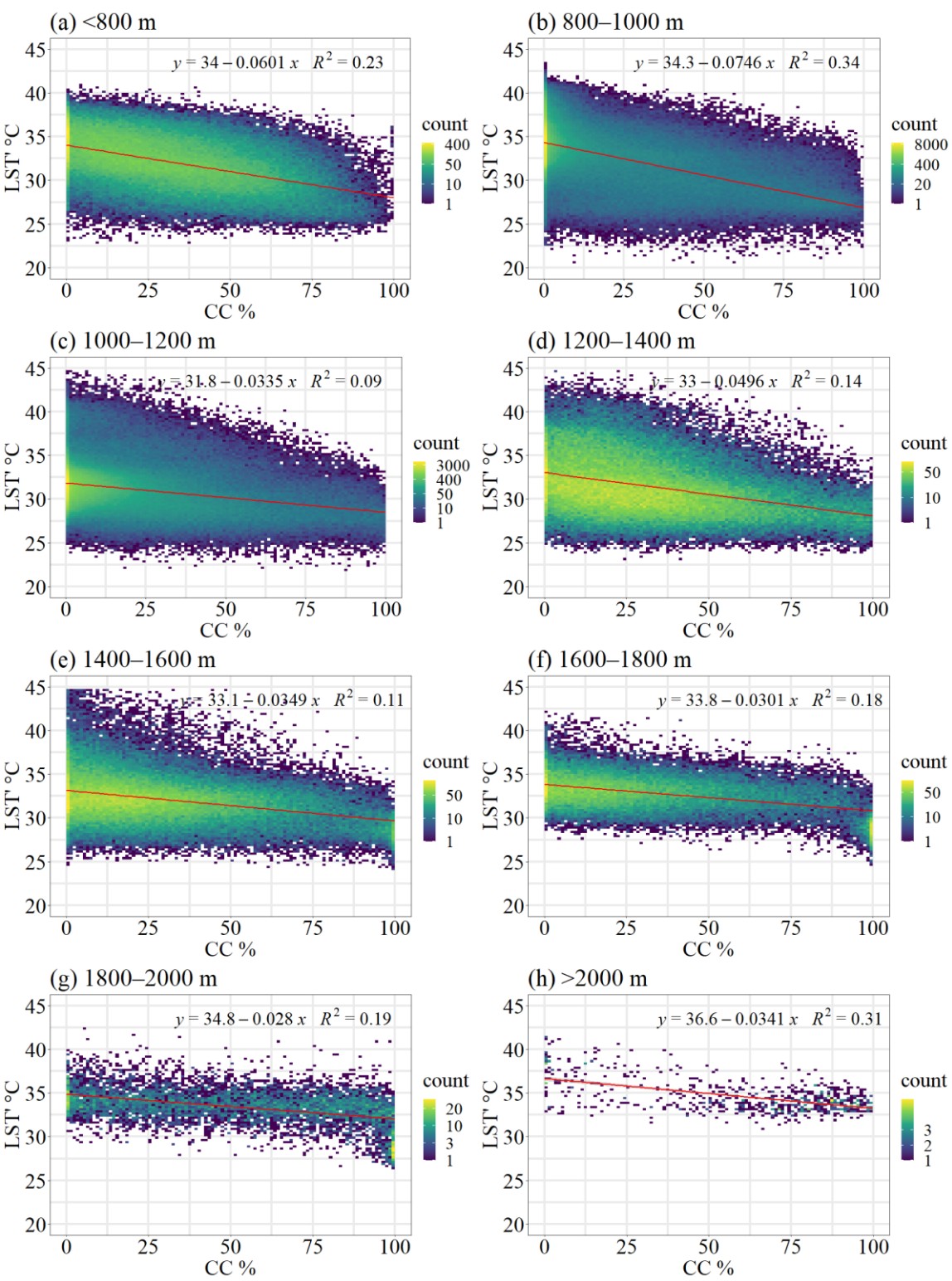


**Figure 7:** Density plots of topographically corrected land surface temperature (LST') and canopy cover (CC) percentage

in eight elevation classes, with regression line. a) below 800 m. b) 800–1000 m. c) 1000–1200 m. d) 1200–1400 m. e)

1400–1600 m. f) 1600–1800 m. g) 1800–2000 m. h) above 2000 m.


## 4. Discussion

High CC decreased near-ground mean temperatures on average by 5.2 °C compared to open land, depending on measurement height. The difference was even greater in temperature maxima, which has been reported to be the case also by De Frenne et al. (2019) and Belsky et al. (1989). Temperature and CC had a linear relationship, pointing out that closed CC was not needed for a substantial cooling effect.

$T_{surface}$ was affected the most by CC. Despite the measurement height of $T_{surface}$ being only 13 cm below $T_{air}$, the effect of CC was notably weaker in $T_{air}$, which is in line with previous studies. For example, Davis et al. (2019) report that the effect of CC was weaker at 2 m than at 10 cm height, while in De Frenne et al. (2019) temperature offset between forest and open land was the greatest close to the ground. In Belsky et al. (1989), soil temperature was the least affected by CC. Luyssaert et al. (2014) compared air temperature and LST and report that the temperature of the planetary boundary was less affected than LST by the removal of forest cover.

Macroclimate affected the magnitude of the cooling: based on the temporal data from the microclimate sensors, during the cooler days of overcast conditions, CC's cooling effect was smaller. Additionally, the temperature differences between low and high CC sites were smaller during these days. In the case of LST, elevation impacted the cooling effect: above 1000 m, the cooling effect decreased by approximately 50 % to that of the lowlands. It can be concluded that trees' importance in controlling temperatures increases in hotter environments. The finding is meaningful, because agricultural expansion on the cost of woody vegetation cover in the area is predicted to take place predominantly in the lowlands (Erdogan et al., 2011; Maeda et al., 2010), where the temperatures are very high. Increasing tree cover on farmlands could thus be of considerable benefit in decreasing local temperatures.

Our finding is in parallel to findings by Zeng et al. (2021), who reported an elevational effect of deforestation on temperatures in Albertine Rift Mountains: the warming effect of deforestation decreased with elevation and disappeared at elevations above 3000 m. This phenomenon resembles the latitude-dependent effect of forests on temperatures: in tropical areas, there is more cooling, while boreal forests cause more warming (Lee et al., 2011; Li et al., 2015). Plant evapotranspiration rates are relative to the solar radiation, ambient temperatures and water balance (Geiger, 1980; Allen et al., 1998; Davis et al., 2019), decreasing the demand for evapotranspiration in low temperatures caused by elevational lapse rate or cool weather conditions. During clear weather, canopies absorb and reflect most of the incoming solar radiation creating cooler conditions in the understory together with evapotranspiration, whereas cloud cover causes a total reduction in the incoming short-wave radiation (Geiger, 1980; De Frenne et al., 2021). Moreover, while the

evapotranspirative cooling mostly offsets warming caused by canopy albedo, in high elevations the albedo effect stays
constant and evapotranspiration decreases (Zeng et al., 2021).
The impact of CC on microclimate was different on different days, and is likely to vary during different times of the year
(Davis et al., 2019; De Frenne et al., 2021). We expect this to be the case with LST as well. For instance, Maeda and
Hurskainen (2014) found that land cover's influence on LST in Mount Kilimanjaro varied seasonally and diurnally, and
the effect was dependent on elevation. Our LST estimation was only a snapshot for July 4, 2019, a sunny almost cloud-
free day, and does not represent the year-round situation experiencing two rainy seasons, which are cloudy. In the hills,
cloudy and misty conditions are experienced throughout the year (Helle, 2016; Räsänen et al., 2018). A time series
comparing the cooling effect of CC over seasons and several years is an interesting future research topic, as the TOMST
sensors remained in the 19 field plots. Interesting would also be to model the sunshine hours every day in the locations
of the TOMST sensors using the hemispherical photography, in order to assess how many hours of the day the tree cover
causes shadows over the sensor.
Canopies control the thermal environments of forests to a high extent (De Frenne et al., 2019; Davis et al., 2019), which
was reaffirmed in this study. Therefore, CC can mitigate large-scale macroclimate warming (De Frenne et al., 2019). An
increase of 2 °C of the global temperature as a consequence of enhanced greenhouse effect can have detrimental impacts
on the most vulnerable ecosystems (IPCC, 2018). Because the time span of local changes in temperatures due to LULCC
is much shorter than in the global climate change, the regional and local consequences can be of even higher magnitude
(Potter et al., 2013; Chen et al., 1999). Due to the debts of species' adaptation capabilities to climate warming (Zellweger
et al., 2020), changes in the microclimate temperatures may be fatal for flora and fauna occupying narrow thermal niches.
This may further impact biodiversity and consequently the crucial ecosystem services provided by forests that take place
close to ground surface (Chen, et al., 1999; Zellweger et al., 2020).
Forest fragmentation decreases the ability of tropical forests to mitigate climate change (Ewers and Banks-Leite, 2013),
but on regional scale even small forests have an impact on LST (Mildrexler et al., 2011). Our results from the linear
models revealed that TOF had the same effect on local temperatures as forests despite the smaller magnitude, and could
hence help in conserving biodiversity. For instance, Mendenhall et al. (2016) found that in Costa Rica farm trees increased
the number of tree and plant species. Most of the CC in Taita Hills comprises of TOF occurring on farms and human
settlement. Sites with agroforestry trees and moderate CC were already experiencing both lower mean and maximum
temperatures than the open sites.
The importance of TOF is receiving more attention (Kuyah et al., 2019; Skole et al., 2021), and in Taita Hills, Pellikka et
al. (2018) reported an addition in carbon stocks since 2003. The Agriculture (Farm Forestry) Rules of 2009 requires that
at least 10 % forest cover should be left or planted on farms. Based on our results, this 10 % CC makes a significant
difference in temperatures (-0.5 °C in mean and -1 °C in maximum temperatures; -0.5 °C in LST). Soil and air
temperatures impact crop productivity, and furthermore, the fog deposit captured by trees brings more water to plants. In
general, increasing temperatures make plant growth more efficient, but this is the case only as long as the increase occurs
within the thermal limits of the plant's tolerance (Muimba-Kankolongo, 2018). As extreme heat and precipitation events
are becoming more common with climate change (MoALF, 2016; IPCC, 2018), the negative effects of warming will
become notable in sub-Saharan Africa. This further threatens the food security, and especially the most common crop,
maize, which is one of the most vulnerable crops in terms of climate change in Africa (Cairns et al., 2013; Adhikari et al.,
2015). Forests of Taita Hills contribute to the food security by capturing atmospheric moisture as fog deposit and storing
the water providing water for farms in the foothills and lowlands (Pellikka et al., 2013; Helle, 2016). In addition to dew
capture, agroforestry has shown to contribute to improved soil moisture (Rhoades 1995; Siriri et al., 2013), hydraulic
conductivity (Nyamadzawo et al., 2003, 2007) and water storage (Makumba et al., 2006; Nyamadzawo et al., 2012).
The pressure on tropical forests in sub-Saharan Africa is caused by many reasons, fuelwood collection being significant
(Abdelgalil, 2004; Zschauer, 2012), which could be mitigated by increasing the tree cover on farms (Unruh et al., 1993;
Iiyama et al., 2014; Chakravarty et al., 2019). The results of this study further encourage to increase tree cover, particularly
in the lowland farms, as a strong potential way to fight the negative effects of climate change. Nevertheless, water is
scarce especially in the lowland areas, and trees' vast need for water must be taken into account. The phenomenon is
paradoxical, because trees improve the water cycle, in general, but are consumes high amounts of water (Ong et al., 2006).
Water balance also affects the temperature buffering capacity of trees (Davis et al., 2019). In areas with water scarcity,
the competition for water resources between crops, animals and people may be a limiting factor in the adoption of
agroforestry practices. One solution in the hot lowlands is dew collection, but it would require a tree cover or other
surfaces to capture the humidity. In Tuure et al. (2019), artificial surfaces produced at best 0.1 liter per day and 25 liters
in a year water from morning dew.
This study was limited to a short time span and a small sample size in microclimate study sites, which makes it susceptible
for uncertainties associated with temporal and spatial variability. Topographic correction was applied on the microclimate
data and was calculated based on elevation only. The small amount of observations did not allow for calculating the
impact of the aspect, which is expected to exist based on the LST analysis. Due to accounting for the effect of topography,
both microclimate and LST estimates did not represent the true values recorded, but made the temperatures comparable
by CC.
In terms of LST, as has been documented in several studies, spaceborne TIR remains an uncertain method for accurate
LST retrieval (Simó et al., 2018; Li et al., 2013). After all, LST is an indirect measurement and the results of complicated
mathematical processing requiring knowledge of several components, where error in any of them causes inaccuracies in
LST (Simó et al., 2018). We calculated LST using the SC method by Jiménez-Muñoz and Sobrino (2004) due to the stray
light problem in Landsat 8 TIRS band 11. While using only one thermal channel for the estimation of LST exposes a high
possibility of inaccuracy, band 10 is more suitable for the SC method than band 11 because of higher atmospheric
transmissivity (Jiménez-Muñoz et al., 2014). The main sources of error in SC are estimation of atmospheric water vapor
content and LSE. LSE is determinant in the correct LST retrieval, yet highly difficult to measure and prone to error. Water
vapor, in turn, can be highly spatially variable, and should be retrieved preferably from satellite data rather than pointwise
weather station data (Ndossi and Avdan, 2016). Jiménez-Muñoz et al. (2014) report that water vapor content higher than
3 g cm$^{-2}$ causes unacceptable inaccuracy: in this study, the water vapor content was 1.7 g cm$^{-2}$, which decreases the
possible error. Wang et al. (2019) conclude that the SC is a valid method for Landsat 8 processing and produces results
on accuracy high enough for most purposes; Ndossi and Avdan (2016) found that SC was the second best algorithm for
the retrieval of Landsat 8 LST. SC has been applied successfully also by for example He et al. (2019). Moreover, in dense
canopies the signal constitutes mostly of the upper canopy (Bense et al., 2016; Zellweger et al., 2019), and previous
studies have not so far demonstrated LST's relationship with understory conditions. We showed how LST provided
consistent results with particularly $T_{surface}$ and $T_{air}$. Therefore, this study contributed to clarifying the relationship of upper
canopy and the understory.
Our study provided information about a topic of which importance has only recently been recognized (De Frenne et al.,
2013; Jucker et al., 2018; Davis et al., 2019; Zellweger et al., 2020). Research and modelling of climate change
implications on microclimate cannot rely on observations from weather stations with low spatial resolution, but need data
that represent the microclimatic conditions relevant for most ecosystem functions (Potter et al., 2013). Previous research
about vegetation and LST have been often conducted at much lower spatial resolutions and applied less accurate
topographic correction (Li et al., 2015). Furthermore, the effect of trees on climate is usually studied solely based on
comparison between forest and open land (De Frenne et al., 2019), neglecting the intermediate canopies and their
significance, despite of the fact that human activity focuses mostly in areas with TOF. We used microclimate data
covering a CC gradient and satellite-derived LST data combined with a DEM of 30 m acquired with ALS over the versatile

study area. While establishing field observation networks with wide spatial coverage remains a challenge, our results showed that LST can be used as a proxy for assessing the impacts of CC on microclimate.

Future research should further investigate the contribution of varied factors to microclimate. For example, since all trees are not of equal benefits in agroforestry, more studies could be targeted to the comparison of different agroforestry species' cooling potential as well as the potential of plantation forests. Including soil moisture, air temperature and comprehensive field plot networks under different canopy structures in the future analyses should broaden the knowledge about trees' role in mitigating and adapting to climate change.

### 5. Conclusions

Our results demonstrate a consistent but heterogeneous influence of canopy cover on the microclimate of highly diverse tropical ecosystems. Daytime temperatures correlated inversely with canopy cover, the effect being strongest on surface temperatures. In hotter environments, the difference between sites of high and low canopy cover became most notable. The cooling effect did not exist only with high canopy cover, but even intermediate canopy cover and trees outside forest buffered the hottest temperatures. Our results thus provide robust evidence that any efforts in the direction of preserving, restoring or increasing vegetation cover can have a substantial impact in creating more stable and cooler microclimates. Satellite-based land surface temperature was a suitable proxy for assessing microclimatic variables surface- and near-ground temperatures, particularly in heterogeneous regions, where the network of field measurements cannot cover the spatial microclimate variability.

This study provided valuable information about the potential of trees in climate change adaptation and mitigation in tropical environments. As the effect of canopy cover on microclimate increased at lower elevations and during hot days, our results indicate that warmer and drier regions are likely to benefit the most from trees.

## Appendix A. Method for hemispherical photography

We took hemispherical photographs at every microclimate sensor site. The camera in use was Nikon D5000 DSLR and the lens Sigma 4.5 mm F2.8 EX DC HSM Circular Fisheye. The camera was attached to a tripod during the taking of photographs. We took photographs at two different heights: the lowest possible tripod adjustment to be as close to the actual sensor level as possible, which was around 60 cm, and at eye-level around 130 cm. We took photographs at eye-level also to every intercardinal direction 15 meters away from the sensor. The camera was adjusted looking upward with the top of the camera pointing north. Two images at every height and direction were taken with different settings: first image on Program mode with automatic aperture and shutter speed, and the second on Manual mode with the rest of the settings staying the same as in picture one, except shutter speed was reduced to half of the first mage. The ISO value was set as constant 500. The purpose of the smaller shutter speed was to reduce the impact of light conditions that were not optimal, meaning direct sunlight that causes overexposure of images which in turn makes them difficult to analyze. Optimally, the photographs should be taken under constant cloud cover or at the dawn or dusk (Pellikka et al., 2000), however due to the timetable, waiting for better light conditions at some sites was not possible, thus some images were overexposed.

We analyzed the hemispherical photographs in the software Hemisfer (WSL; version 2.2) (Schleppi et al., 2007; Thimonier et al., 2010). From the two images, we used the less exposed one in the analysis. For the calculation of canopy cover, we used the images taken from eye-level, because they were more comparable to the ALS-based canopy cover, and the photographs in cardinal directions were all taken at eye-level. We classified the image pixels to sky and canopy by determining a threshold value to separate dark and light pixels in the image. For most images, we used the automatic threshold method by Nobis and Hunziker (2005). In the case of some images, the algorithm clearly produced errors due to overexposure and direct sunlight, therefore the algorithm by Ridler and Calvart (1978) was applied, or a manual threshold was determined. We used only the blue band in the analysis, apart from photographs where the classification was failing and using all the bands produced the best result (Heiskanen et al., 2015a). The gamma correction was $\gamma = 2.2$. Only the zenith angle range of $0\text{-}15°$ was analyzed, because errors in canopy cover accuracy increase with larger angles (Paletto and Tosi, 2009). We computed canopy cover by calculating an average of 1-gap fraction of the five measurements, and this gave a plot-wise canopy cover (Heiskanen, et al., 2015b). Finally, we compared the canopy cover retrieved from hemispherical photography and ALS using Pearson's correlation and a Student's t-test. The mean of differences was 0.89 and was not statistically significant.

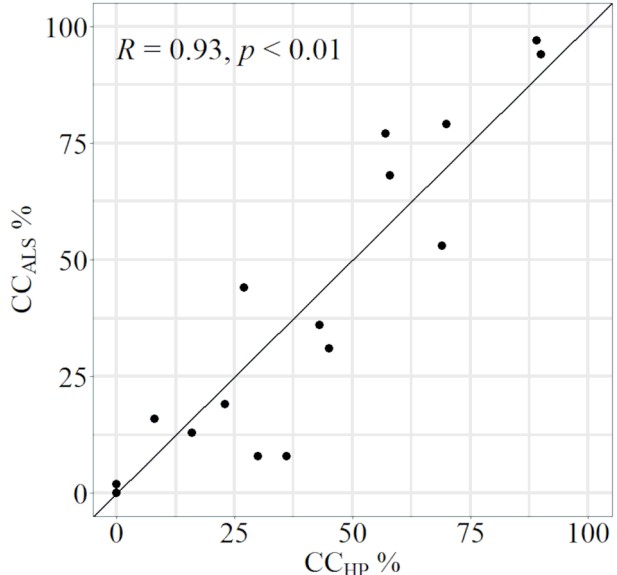

$R = 0.93, p < 0.01$


**Figure A1.** Comparison of canopy cover (CC) percentage retrieved from airborne laser scanning (ALS) and hemispherical
photography (HP), with line of identity.
**Appendix B. Results of the linear regression models of land surface temperature**

| Predictor | Model | Coef | Std. Error | T-Value | P-Value |
|---|---|---|---|---|---|
| Constant | 1 | 44.79 | 0.013 | 3324.0 | <0.001* |
| | 2 | 44.24 | 0.019 | 2300.9 | <0.001* |
| | 3 | 46.71 | 0.018 | 2580.3 | <0.001* |
| | 4 | 44.08 | 0.021 | 2130.9 | <0.001* |
| Elevation | 1 | -0.013 | 0.000 | -1241.4 | <0.001* |
| | 2 | -0.011 | 0.000 | -577.2 | <0.001* |
| | 3 | -0.015 | 0.000 | -954.6 | <0.001* |
| | 4 | -0.012 | 0.000 | -592.3 | <0.001* |
| Slope | 1 | -4.061 | 0.018 | -220.0 | <0.001* |
| | 2 | -3.806 | 0.018 | -214.9 | <0.001* |
| | 3 | -3.723 | 0.018 | -202.3 | <0.001* |
| | 4 | -1.545 | 0.054 | -28.534 | <0.001* |
| Canopy cover | 1 | -0.050 | 0.000 | -419.0 | <0.001* |
| | 2 | -0.068 | 0.000 | -449.1 | <0.001* |
| | 3 | -0.109 | 0.000 | -274.7 | <0.001* |
| | 4 | -0.068 | 0.000 | -452.4 | <0.001* |
| NE | 1 | 0.177 | 0.011 | 16.0 | <0.001* |
| | 2 | 0.084 | 0.010 | 8.1 | <0.001* |
| | 3 | 0.157 | 0.011 | 14.3 | <0.001* |
| | 4 | -0.148 | -0.016 | -9.4 | <0.001* |
| E | 1 | -0.030 | 0.010 | -29.0 | <0.001* |
| | 2 | -0.428 | 0.010 | -44.6 | <0.001* |
| | 3 | -0.352 | 0.010 | -34.7 | <0.001* |
| | 4 | -0.452 | 0.016 | -32.4 | <0.001* |
| SE | 1 | -1.447 | 0.010 | -140.0 | <0.001* |
| | 2 | -1.509 | 0.010 | -155.6 | <0.001* |

| | | | | | |
|---|---|---|---|---|---|
| | 3 | -1.529 | 0.010 | -149.3 | <0.001* |
| | 4 | -1.178 | 0.014 | -85.4 | <0.001* |
| S | 1 | -2.095 | 0.011 | -189.4 | <0.001* |
| | 2 | -2.132 | 0.010 | -205.2 | <0.001* |
| | 3 | -2.186 | 0.011 | -199.4 | <0.001* |
| | 4 | 1.543 | 0.014 | -107.3 | <0.001* |
| SW | 1 | -2.441 | 0.011 | -230.0 | <0.001* |
| | 2 | -2.554 | 0.010 | -256.0 | <0.001* |
| | 3 | -2.527 | 0.011 | -240.1 | <0.001* |
| | 4 | -1.820 | 0.014 | -130.2 | <0.001* |
| W | 1 | -2.293 | 0.010 | -219.5 | <0.001* |
| | 2 | -2.254 | 0.010 | -229.9 | <0.001* |
| | 3 | -2.332 | 0.010 | -225.5 | <0.001* |
| | 4 | -1.554 | 0.014 | -109.2 | <0.001* |
| NW | 1 | -1.380 | 0.011 | -126.8 | <0.001* |
| | 2 | -1.205 | 0.010 | -117.9 | <0.001* |
| | 3 | -1.379 | 0.012 | -127.9 | <0.001* |
| | 4 | -0.883 | 0.015 | -58.5 | <0.001* |
| 1000-1500 m | 1 | . | . | . | . |
| | 2 | -2.667 | 0.008 | -346.9 | <0.001* |
| | 3 | . | . | . | . |
| | 4 | -2.645 | 0.008 | -346.8 | <0.001* |
| >1500 m | 1 | . | . | . | . |
| | 2 | -2.030 | 0.018 | -111.2 | <0.001* |
| | 3 | . | . | . | . |
| | 4 | -1.875 | 0.018 | -103.5 | <0.001* |
| Canopy cover: 1000–1500 m | 1 | . | . | . | . |
| | 2 | 0.031 | 0.000 | 149.7 | <0.001* |
| | 3 | . | . | . | . |

| | | | | | |
|---|---|---|---|---|---|
| | 4 | 0.031 | 0.000 | 151.2 | <0.001* |
| Canopy cover: >1500m | 1 | . | . | . | . |
| | 2 | 0.028 | 0.000 | 120.7 | <0.001* |
| | 3 | . | . | . | . |
| | 4 | 0.038 | 0.000 | 122.5 | <0.001* |
| Elevation: canopy cover | 1 | . | . | . | . |
| | 2 | . | . | . | . |
| | 3 | 0.00005 | 0.000 | 156.3 | <0.001* |
| | 4 | . | . | . | . |
| Slope: NE | 1 | . | . | . | . |
| | 2 | . | . | . | . |
| | 3 | . | . | . | . |
| | 4 | 0.798 | 0.062 | 11.8 | <0.001* |
| Slope: E | 1 | . | . | . | . |
| | 2 | . | . | . | . |
| | 3 | . | . | . | . |
| | 4 | -0.144 | 0.060 | -2.387 | 0.017 |
| Slope: SE | 1 | . | . | . | . |
| | 2 | . | . | . | . |
| | 3 | . | . | . | . |
| | 4 | -2.014 | 0.061 | -33.1 | <0.001* |
| Slope: S | 1 | . | . | . | . |
| | 2 | . | . | . | . |
| | 3 | . | . | . | . |
| | 4 | -4.045 | 0.067 | -60.0 | <0.001* |
| Slope: SW | 1 | . | . | . | . |
| | 2 | . | . | . | . |
| | 3 | . | . | . | . |
| | 4 | -0.943 | 0.063 | -78.1 | <0.001* |

| | | | | | |
|---|---|---|---|---|---|
| Slope: W | 1 | . | . | . | . |
| | 2 | . | . | . | . |
| | 3 | . | . | . | . |
| | 4 | -3.918 | 0.060 | -64.8 | <0.001* |
| Slope: NW | 1 | . | . | . | . |
| | 2 | . | . | . | . |
| | 3 | . | . | . | . |
| | 4 | -1.963 | 0.065 | -30.4 | <0.001* |


**Table B1:** Summary of regression coefficients in the analysis of land surface temperature (LST) from the four models
tested. * indicates statistical significance.

**Data and code availability**

The data and scripts presented in this study are available on request from the author (I.A.).

**Author contribution**

Conceptualization, I.A., E.M., J.H. and P.P.; data curation, I.A.; formal analysis, I.A., E.A.; funding acquisition, P.P.; investigation, I.A., methodology, I.A, E.M., J.H., E.A. and P.P.; project administration, E.M. and P.P.; resources, software, I.A.; supervision, E.M, J.H. and P.P.; validation, I.A., visualization, I.A., writing—original draft preparation, I.A.; writing—review and editing, IA., E.M., J.H. and P.P. All authors have read and agreed to the published version of the manuscript.

**Declaration of Competing Interest**

The authors declare that they have no conflicts of interest.

**Funding**

This study was conducted as part of Smartland project (Environmental sensing of ecosystem services for developing a climate-smart landscape framework to improve food security in East Africa, decision no. 31864) funded by Academy of Finland, and ESSA project (Earth observation and environmental sensing for climate-smart sustainable agropastoral ecosystem transformation in East Africa) funded by European Commission DG International Partnerships DeSIRA programme (FOOD/2020/418-132). Eduardo Maeda was funded by the Academy of Finland (decision numbers 318252 and 319905).

**Acknowledgements**

We would like to acknowledge Agnes Mwangombe, Ali Ndizi, Mrs. Mwamburis, Mrs. Nyatta, Cathrine Mwakesi, Simon, Moses Onyimbo and Dalmas moka secondary school, Jason Collette and Teita Sisal Estate, St. Mary's Teachers' Training College, and Taita Taveta University Ngerenyi campus for allowing us to conduct this research on their properties. We also thank Taita Research Station of the University of Helsinki for logistical support during the field wok campaign. Special thanks to Mwadime Mjomba for assistance during the field work. We acknowledge Matti Räsänen for the provision of weather station data and Hari Adhikari for the canopy cover data. We also want to thank the two anonymous reviewers for their comments and suggestions to improve the manuscript.

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
