# Peer review of "Strong influence of trees outside forest in regulating microclimate"

_Biogeosciences, 2021_

## Referee Comment (RC2)

Review on Aalto et al. *Strong influence of trees outside forest in regulating microclimate of intensively modified Afromontane landscapes*

This research topic is interesting and important, and data analysis is described clearly. However, the results interpretations and conclusion parts need to be improved. The main finding of the research is that canopy cover reduces the microclimate temperature, and this cool-down regulation decreases with elevation. For me, the former finding (i.e. canopy cover down-regulates microclimate) is not very novel, which has been reported in many earlier studies (e.g. Zellweger et al., 2020; Fig. S2 and S5). In contrast, the latter finding (i.e. the influence of canopy cover on microclimate is regulated by the elevation or ambient temperature) is novel and very interesting, but the authors did not say much about this result. I think this part should be emphasized and enhanced descriptions are needed (e.g. discuss that what are the potential mechanisms? refer to Zeng et al., 2019). Meanwhile, some other things also should be clarified, such as macroclimatic temperature also largely affects the microclimate, how did you address or consider this issue in your work? The effect of canopy cover on microclimate and satellite LST is very different in mechanistic (please see the general comments and specific comments).

General comments

1) Macroclimatic temperature also largely affects the microclimate (e.g. Fig. 1c in Zellweger et al., 2020), how did you address or consider this issue in your work? Or you think the topography has included the macroclimatic influence? Please clarify

2) The effects of canopy cover on microclimate and LST are different from the mechanistic perspective. The effect on microclimate is mixing and shading. The effect on LST is the temperature difference between vegetation canopy and background surface temperature (e.g. soil). Microclimate and LST just have a similar negative correlation with canopy cover. Thus, should take cautions when expanding the LST pattern or findings to microclimate, and *vice versa.* Please clarify

3) The effect of canopy cover on microclimate is regulated by ambient temperature (and the effect on LST is regulated by elevation) are very interesting findings. Maybe better to provide more evidence and descriptions. Such as the across diurnal timescales, ambient temperature change a lot (from low temperature to high temperature), how CC on microclimate change, give the relationship panels (e.g. x-axis mean temperature, y-axis: CC effect on microclimate, I guess there is a positive correlation between them, i.e. higher temperature, higher CC effect).

4) The analysis of LST should be more rigorous. For Landsat 8 satellite, although you have taken some corrections, only one thermal band could be used, and the uncertainty remained large. Maybe you should add the comparison between MODIS LST (three thermal bands are available, 1km resolution, which has also been widely used) and Landsat 8 LST to validate the accuracy of Landsat 8. Meanwhile, you only analyze the LST at 10:30, the analysis of 13:30 LST is also necessary, since it corresponds to the near-maximum temperature within a day, and the effect of CC on LST may be larger at 13:30 than that at 10:30.

5) The results or visualization about the effect of canopy cover on microclimate variability should be improved (Line:191-195).

Specific comments:

Line 22-23: "......vary strongly with elevation and ambient temperatures". It's unclear to me, what're the ambient temperatures refer to? (if it is macroclimatic air temperature? I guess you did not measure this indicator)

Line 24: what do the macroclimatic conditions mean?

Line 78: how to quantify the stability of microclimates?

Line 86-87: please provide the LAT, LON location information, although this could be interpreted from the Fig. 1

Line 103-107: this information would be better to be assimilated into the introduction part

Line 122-124: Please provide more information about microclimate sensors installed environments, they will affect collected microclimate data. For example, at low CC sites, the sensor was installed shadow area or sunlit area, and there are large temperature differences between sun and shade, meanwhile sunlit or shadow saturation also changes with sun angles, how did you consider these issues?

Line 129-131: macroclimate also largely affects microclimate (e.g. Fig. 1c in Zellweger et al., 2020), how do you consider this issue or how do you eliminate the influence of CC on microclimate? Meanwhile, whether the elevation and CC independent (independent assumption of the linear model)?

Line 134: please add the values.

Line 137-141: please clarify whether the mismatch of the collection date between ALS data and microclimate data could influence your research results, or evaluate its uncertainty?

Line 159: how do you calculate the land surface emissivity?

Line 170-174: I am worried about this correction: this method could be regarded as three independent steps: the first step is removing elevation influence by minus dTh; the second step is removing slope influence by minus dTh; the third step is removing aspect influence by minus dTa. It makes sense if these three factors are independent, if they are highly correlated, this method will over-correct the LST?

Line 191: 'affected also' → 'also affected'; please give the full definition of SD when it first appears

Line 191-195: where are these results from? I guess maybe from the Figure, this information may not be obtained by readers directly. please clarify.

Line 258-261: what's the underlying season the cooling impact of CC decease with elevation? Table 4 may be moved into supplementary

Line 286-287: Not clear to me, the sensible flux of canopy means the sensible heat exchange between canopy surface temperature and surrounding air temperature. Only when canopy temperature equals surrounding air temperature, we can say no sensible effect.

Reference:

Zellweger, F., De Frenne, P., Lenoir, J., Vangansbeke, P., Verheyen, K., Bernhardt-Römermann, M., ... & Coomes, D. (2020). Forest microclimate dynamics drive plant responses to warming. Science, 368(6492), 772-775.

Zeng, Z., Wang, D., Yang, L., Wu, J., Ziegler, A. D., Liu, M., ... & Wood, E. F. (2021). Deforestation-induced warming over tropical mountain regions regulated by elevation. Nature Geoscience, 14(1), 23-29.

---

## Author Response (AR1)

**Author's response**

Aalto et al.: Strong influence of trees outside forest in regulating microclimate of intensively modified Afromontane landscapes

**Reviewer #1**

**General comments:**

This paper aims to examine the effect of canopy cover on microclimate in an intensively modified Afromontane landscape in Taita Taveta, Kenya. The authors studied microclimate sensors under different canopy covers, and land surface temperature (LST) from Landsat 8 thermal infrared sensors and combined these data with high-resolution airborne laser scanning data to disentangle combined effects of topography and canopy cover on microclimate. This is an interesting comparison of temperatures across canopy cover changes, forest types, and elevational gradients to help understand thermal regulation by forests and microclimates that buffer species and local climate against warming.

A strength of the study is that it utilizes multiple temperature datasets (in situ and remote sensing) and that it includes study sites along an elevational gradient with CC changes. The study shows a strong negative relationship between canopy cover and temperature, with the strongest effect on maximum temperature. Results are well reported with good figures. The main weakness of the paper is that it does not carefully describe the different physical measurements, what they actually measure, and how they related and interact. The link between microclimate and LST is not clearly established.

**Reply:** We would like to thank the reviewer for the excellent comments and suggestions. We have addressed each of the comments and made the suggested edits to the manuscript. Especially, we have tried to improve the link between microclimate and LST and explain these concepts more thoroughly. We believe the manuscript has greatly improved after the revision. Please find below detailed replies to all comments.

**Main comments**

Introduction

1. The flow and clarity of the manuscript could be improved. The second paragraph of the Introduction attempts to cover a large range of topics, spatial scales, and physical measurements and is confusing as written. It could be improved with better explanations and breaking it into two separate paragraphs. One paragraph could focus on forests, TOF, the goods and services they provide, and the land use factors impacting these systems. The second paragraph could focus on microclimates and improved description of other temperature measurements and how they relate.

Line 40-41 describes the values of Forests and TOF as "vital ecosystem services" which I think implies that they are source of many goods and services to humans (ie. That next sentence is redundant). When you say that forests and TOF are "also a source of goods for humans," are you actually referring to uses that degrade forests, such as logs for building or firewood gathering? This should be clarified because it can conflict with the ecosystem services you have just described. The tension between what forests provide when they are left standing, and how they are used for raw materials sets the stage for the existing condition of your study area.

On lines 97 to 102 of the Methods you describe this. But I think this is important background on TOF and helps readers understand why TOF are important (e.g. carbon storage and biodiversity). I recommend moving these lines to the Introduction to complete the TOF paragraph.

**Reply:** The second paragraph of the Introduction was split into two separate paragraphs as suggested. Text about ecosystem services in lines 40-41 was clarified. We expanded the description of TOF to highlight its importance.

2. Line 44-53: I recommend a new paragraph starting with the definition of microclimate. The description of microclimates is provided at a scale of centimeters to meters (line 45) which does not always seem congruent with your application of the term microclimate, and the spatial focus of the study. It seems that you are interested in microclimatic buffering capacity of forests at much larger scales? This may in part be a problem with definitions as microclimates can also be defined at very localized scales to describe unique thermal niches within context of the overall forest thermal environment. This seems different than the climatic conditions below forest canopies in general sometimes referred to as "microclimatic buffering". I recommend defining microclimate so that it incorporates the spatial domain of your study question.

For example, the De Frenne study states "…the local temperature experienced by living organisms (referred to as the ('microclimate')… While quite general, this definition includes the thermal regulation of forests across scales.

**Reply:** We created a new paragraph as suggested, containing the definition of microclimate and microclimatic buffering. We believe that a misunderstanding of the definition of these two terms might have caused confusion. The microclimate, as defined in our introduction, is our main interest, and not microclimatic buffering. We defined microclimatic buffering as the difference between the macroclimate and microclimatic conditions under the canopy (Ewers & Banks-Leite 2013). We therefore improved the manuscript to avoid this misunderstanding.

Ewers, R. M., and Banks-Leite, C.: Fragmentation Impairs the Microclimate Buffering Effect of Tropical Forests, PLoS One, 8, e58093, 2013

3. Lines 48-45: The authors compare understory microclimate variability to continental scale studies of spatial variability in LST measurements. LST is canopy temperature, a different physical measurement that indicates partitioning of solar radiation driven by transpirational cooling. I think you should add a sentence recognizing the differences in these physical measurements and that you are inferring a relationship between radiometric surface temperature and understory temperatures especially since the relationship between these various temperature measurements are not well understood.

**Reply:** We thank the reviewer for the comment. We edited the sentence to avoid misunderstanding and added a more detailed description of LST in the paragraph introducing it.

4. Line 56: Numerous studies have examined forests and moisture gradients using LST and thermal imagery from fine to moderate scales.

Scherrer, D. M. K-F. Bader, and C. Korner. 2011. Drought-sensitivity ranking of deciduous tree species based on thermal imaging of forest canopies. Agricultural and Forest Meteorology, 151, 1632-1640.

Kim, Y., Still, C.J., Hanson, C.V., Kwon, H., Greer, B.T. & Law, B.E. (2016). Canopy skin temperature variations in relation to climate, soil temperature, and carbon flux at a ponderosa pine forest in central Oregon, Agricultural and Forest Meteorology, 226, 161-173.

Mildrexler, D.J., Yang, J.Z., & Cohen, W.B. 2016. A Forest Vulnerability Index Based On Drought and High Temperatures. Remote Sensing of Environment, 173, 314-325.

**Reply:** We agree with the reviewer that studies using LST to examine forests are abundant. However, studies focusing on the microclimate of tropical areas are still few: weather station data is still commonly used to study climate change effects. We have edited the sentence to highlight this.

5. Line 56-60: Consider that lines 103-107 of the Methods would make more sense added here. You are describing the climate mitigation potential of forests and TOF, so this would be a good place to highlight the projections for your study area that make this so important.

**Reply:** We thank the reviewer for the suggestion and have moved the lines from Methods to the Introduction.

6. 62-72: The background on LST and CC should also note the well-described negative relationship between LST and vegetation density. It is key to your study of fractional vegetation coverage (see references below for examples). This addition would be a good transition into the paragraph starting at line 68 that describes some of the commonly used vegetation indices used to study this relationship.

Nemani, R. R., and S. W. Running 1997. Land cover characterization using multitemporal red, near-IR, and thermal-IR data from NOAA/AVHRR, Ecol. Appl., 7, 79–90.

Goward, S. N., Cruickshanks, G. D., & Hope, A. S. (1985). Observed relation between thermal emission and reflected spectral radiance of a complex vegetated landscape, Remote Sensing of Environment, 18, 137-146.

Goward, S. N., & Hope, A. S. (1989). Evapotranspiration from combined reflected solar and emitted terrestrial radiation: Preliminary FIFE results from AVHRR data., Advanced Space Research, 9, 239-249.

**Reply:** We thank the reviewer for the excellent suggestions. We have expanded the introduction of LST and included the references.

Methods and Results

7. Microclimatological field measurements: Is it common in microclimatological field measurements for Tair to be only 15 cm from the ground? It seems that measurements so close to the ground would be influenced by the surface temps. I am accustomed to Tair from weather stations where it refers to temperatures measured 1.5 meters above the ground surface as in:

Oyler, J. W., S. Z. Dobrowski, Z. A. Holden, and S. W. Running, 2016: Remotely sensed land skin temperature as a spatial predictor of air temperature across the conterminous United States. J. Appl. Meteor. Climatol., 55, 1441–1457

**Reply:** Weather stations that measure temperature at 1.5 meters cannot capture the microclimatic conditions. The 15 cm is already common in microclimatic measurements (see example references below). We used the acronym "Tair" to separate it from the two other measurement heights of the

TOMST sensor (Tsoil and Tsurf), so the Tair here is not equivalent to the commonly used Tair of 1.5 m. We have clarified this in the text.

Wild, J., Kopecký, M., Maeck, M., Sanda, M., Jankovec, J., and Haase, T.: Climate at ecologically relevant scales: A new temperature and soil moisture logger for long-term microclimate measurement, Agr. Forest Meteorol., 268, 40–47, 2019.

Maclean, Ilya M. D et al.: On the Measurement of Microclimate, Methods in ecology and evolution, 12, 1397–1410, 2021.

De Pauw, Karen et al.: Forest Understorey Communities Respond Strongly to Light in Interaction with Forest Structure, but Not to Microclimate Warming, The New phytologist, 233, 219–235, 2022.

Doughty, Christopher E et al.: Forest Thinning in Ponderosa Pines Increases Carbon Use Efficiency and Energy Flow From Primary Producers to Primary Consumers. Journal of geophysical research. Biogeosciences, 126, 2021

Macek, Martin et al.: Elevational Range Size Patterns of Vascular Plants in the Himalaya Contradict Rapoport's Rule, The Journal of ecology 109, 4025–4037, 2021.

Vandvik, V., Halbritter, A.H., Yang, Y. et al. Plant traits and vegetation data from climate warming experiments along an 1100 m elevation gradient in Gongga Mountains, China. Sci Data 189, 2020.

8. Lines 150-175. The authors use a single channel method to derive LST as a work around to the stray light problem with TIRS band 11 on board Landsat 8. Can you report the uncertainties associated with this approach?

**Reply:** The uncertainty was addressed very briefly in the discussion, and we agree with the reviewer that it should be mentioned in the methods as well. We have corrected this and added the uncertainty estimate.

9. Figure 5: This is for mean temperatures. Why not show a similar scatterplot sequence for maximum temperatures?

**Reply:** We included standard deviation in the mean temperature figure, and added a similar figure with maximum temperatures:

[Figure]

Discussion

10. The Discussion contains more grammatical problems compared to the rest of the manuscript. I encourage the authors to edit this section for grammatical consistency with the rest of the manuscript.

**Reply:** We thank the reviewer for the feedback and have revised the text for grammatical consistency.

11. Lines 320-322: States that results revealed trees on farms had the same effect on local temperatures as forests… Please refer to the figure or data specifically from the results that supports this finding.

**Reply:** We base this finding on the canopy cover model that showed the linear relationship between CC and temperature. Most of the canopy cover in the area consists of TOF. We have improved the text to clarify this.

12. Lines 323-325: Can you say on average how much cooler these moderately forested TOF sites in agriculture are vs. ag with no TOF?

**Reply:** See reply above.

13. Lines 327-328: This is a great link to local policy. Please state how much the difference in temperature 10% forest cover makes here.

**Reply:** We think that it is an excellent idea to add the numbers to the sentence and did as the reviewer suggested.

14. Lines 334-336: The link between forest and water for agriculture is important. You highlight fog drip here, but what about forests supporting clean water to farms, water retention through increased soil organic matter, and hydraulic redistribution? In other words, try to strengthen this section.

**Reply:** We added a sentence about agroforestry's relationship with water as suggested by the reviewer.

15. Lines 337-338: It is not obvious to me why increasing the tree cover on farms would mitigate pressure on tropical forests from fuelwood collection. I presume that farms are private land, and fuel wood collection is not permitted by the general community, but I do not know. Please explain this.

**Reply:** Thank you for the very relevant comment that deserves clarification. According to Zschauer (2012), large part of the fuelwood in the study area is collected from people's own farms or other areas outside forests, such as bushland. Hence, agroforestry and tree planting in the farms are important for reducing pressure on fuelwood collection from remaining montane forests. The potential of agroforestry to reduce wood harvest pressures in sub-Saharan Africa are also more widely acknowledged (Iiyama et al. 2014).

Iiyama M, Neufeldt H, Dobie P, Njenga M, Ndegwa G, Jamnadass R (2014). The potential of agroforestry in the provision of sustainable woodfuel in sub-Saharan Africa. Current Opinion in Environmental Sustainability, 6, 138-147. https://doi.org/10.1016/j.cosust.2013.12.003

Zschauer K. (2012). Households energy supply and the use of fuelwood in the Taita Hills, Kenya. MSc thesis. University of Helsinki, Faculty of Science, Department of Geosciences and Geography. http://urn.fi/URN:NBN:fi-fe201201311271

16. Lines 349-351: topographic manipulation of the temperatures? Do you mean "accounting for the effect of topography on LST measurements…"?

**Reply:** The reviewer is correct that the formulation could be improved. We have changed this to what the reviewer suggested.

17. 354-355: Can you report an uncertainty estimate for the LST product?

**Reply:** See reply to 8.

18. 355-356: Here you state that LST may not be representative of understory conditions. Wasn't this part of your study? What did you find?

**Reply:** We thank the reviewer for the excellent comment. We have used both microclimatic measurements and LST to study CC and temperature and whether they are telling the same story in our study area. Previous research has not demonstrated the relationship between LST and the understory in dense canopies, and our results have shown that LST can be representative of the understory conditions. We have improved the manuscript to clarify this result.

19. 360: I recommend adding the following citation to this sentence:

Davis, K. T., Dobrowski, S. Z., Holden, Z. A., Higuera, P. E., and Abatzoglou, J. T. (2019a). Microclimatic buffering in forests of the future: the role of local water balance. Ecography 42, 1–11. doi: 10.1111/ecog.03836

**Reply:** We thank the reviewer for the suggestion and have added this reference to the sentence.

20. 369-370: At line 356 you questioned whether LST is representative of understory conditions and provided citations. Here you state that your results showed that LST can be used as a proxy for assessing the impacts of CC on microclimate. Do your results really show this? Or do they show that an increase in canopy cover results in a commensurate decrease in LST?

**Reply:** See reply to 18.

**Other Specific Comments:**

Line 140: delete "laying" so sentence reads "as the plot was outside of the ALS coverage."

**Reply:** We deleted the word "laying".

Line 191: Reword to: CC also affected temperature variability

**Reply:** We reworded the sentence as suggested.

Line 287: In additional to sensible heat flux, could some of the cooling effect be due to transpirational cooling?

**Reply:** We agree with the reviewer that the cooling effect is also due to this. However, the sentence did not mean to separate sensible heat flux and transpiration, and we have reformulated the sentence for more clarity.

298: Change discovery to finding.

**Reply:** We changed the word from "discovery" to "finding".

292: prevalent doesn't make sense here. Please clarify.

**Reply:** We changed "prevalent" to "ambient".

295: "One likely reason…": isn't another likely reason that on cloudy days there is less solar insolation at the surface, which has a disproportionately large effect on warming areas with low CC on sunny days?

**Reply:** We thank the reviewer for the remark and have added a sentence about this in the paragraph.

314: Change extent to magnitude

**Reply:** We changed "extent" to "magnitude".

328: Can simplify sentence to "Soil and air temperatures impact crop productivity,…"

**Reply:** We simplified the sentence by removing "have an".

This research topic is interesting and important, and data analysis is described clearly. However, the results interpretations and conclusion parts need to be improved. The main finding of the research is that canopy cover reduces the microclimate temperature, and this cool-down regulation decreases with elevation. For me, the former finding (i.e. canopy cover down-regulates microclimate) is not very novel, which has been reported in many earlier studies (e.g. Zellweger et al., 2020; Fig. S2 and S5). In contrast, the latter finding (i.e. the influence of canopy cover on microclimate is regulated by the elevation or ambient temperature) is novel and very interesting, but the authors did not say much about this result. I think this part should be emphasized and enhanced descriptions are needed (e.g. discuss that what are the potential mechanisms? refer to Zeng et al., 2019). Meanwhile, some other things also should be clarified, such as macroclimatic temperature also largely affects the microclimate, how did you address or consider this issue in your work? The effect of canopy cover on microclimate and satellite LST is very different in mechanistic (please see the general comments and specific comments).

**Reply:** We would like to thank the reviewer for the comments and suggestions and believe that they have helped us to improve the manuscript tremendously. Particularly, we have improved the discussion of the results based on the reviewer comments, as well as the definitions and descriptions of relevant terms and concepts. We also want to thank for the two great reference suggestions; we have included them in the manuscript.

We have addressed and answered all the comments; please see detailed replies below.

**General comments:**

1) Macroclimatic temperature also largely affects the microclimate (e.g. Fig. 1c in Zellweger et al., 2020), how did you address or consider this issue in your work? Or you think the topography has included the macroclimatic influence? Please clarify

**Reply:** All microclimate data were collected at the same time and plots were relatively close to each other. Therefore, we concluded that the macroclimate was similar in every plot and we would only need to account for topographic differences. We have clarified this in the text.

2) The effects of canopy cover on microclimate and LST are different from the mechanistic perspective. The effect on microclimate is mixing and shading. The effect on LST is the temperature difference between vegetation canopy and background surface temperature (e.g. soil). Microclimate and LST just have a similar negative correlation with canopy cover. Thus, should take cautions when expanding the LST pattern or findings to microclimate, and vice versa. Please clarify

**Reply:** We agree with the reviewer that the mechanics of microclimate and LST are different. We have used both microclimatic measurements and LST to study CC and temperature, and whether they are telling the same story in our study area. Previous research has not demonstrated the relationship between LST and the understory in dense canopies, and our results have shown that LST

can be representative of the understory conditions. We have improved the manuscript to clarify the difference between microclimate and LST, and described our result more carefully.

3) The effect of canopy cover on microclimate is regulated by ambient temperature (and the effect on LST is regulated by elevation) are very interesting findings. Maybe better to provide more evidence and descriptions. Such as the across diurnal timescales, ambient temperature change a lot (from low temperature to high temperature), how CC on microclimate change, give the relationship panels (e.g. x-axis mean temperature, y-axis: CC effect on microclimate, I guess there is a positive correlation between them, i.e. higher temperature, higher CC effect).

**Reply:** Thank you for the excellent suggestion. We added a panel to the figure describing the temporal T in the study period (see figure below), where we plotted CC's cooling effect (regression coefficient) against the dates. The cooling effect varies from approximately 2.5°C to 8 °C, showing how the cooling effect increases during hot days and vice versa.

[Figure]

4) The analysis of LST should be more rigorous. For Landsat 8 satellite, although you have taken some corrections, only one thermal band could be used, and the uncertainty remained large. Maybe you should add the comparison between MODIS LST (three thermal bands are available, 1km resolution, which has also been widely used) and Landsat 8 LST to validate the accuracy of Landsat 8. Meanwhile, you only analyze the LST at 10:30, the analysis of 13:30 LST is also necessary, since it

corresponds to the near-maximum temperature within a day, and the effect of CC on LST may be larger at 13:30 than that at 10:30.

**Reply:** Thank you for the comment. While we recognize the risks of only using Landsat 8 band 10 to calculate LST, the literature suggests it to be an accurate enough method (see for example Wang et al. 2019, He et al. 2019, Yu et al. 2014, Jiménez-Muñoz et al. 2014). The errors reported in previous studies are mostly coming from atmospheric effects. The water vapor content during the taking of the Landsat image was 1.7 g/cm2, which makes the SC method more reliable; Jiménez-Muñoz et al. (2014) conclude that the SC errors become unacceptably high at water vapor contents >3 g/cm2.

Fortunately, the SC method is more accurate for TIR channels that are close to 11 μm than 12 μm due to atmospheric transmittance, which makes channel 10 more suitable for SC method than the use of channel 11 (Jiménez-Muñoz et al. 2014). Jiménez-Muñoz et al. (2014) report an error of 1.5 K; in Ndossi & Avdan (2016) SC and Planck function were the most accurate methods for LST retrieval for Landsat 8.

We find MODIS spatial resolution not great (1 km vs. 30 m) for our study, especially with the heterogeneous topography in the area. In one kilometer, the elevation can vary up to several hundred meters, and consequentially the LST would vary by several degrees based on the elevational lapse rate. Because we are interested in a scale where even individual trees matter to LST, we do not see MODIS to be suitable for our purposes, not even to validate Landsat 8 LST. Unfortunately, there is no Landsat data available at 13.30, but we do agree that it would be very useful and interesting to analyze LST at the time when trees are expected to have the highest cooling effect. We have added a sentence in the discussion about this. May that remain a future research idea.

References:

He, J., Zhao, W., Li, A., Wen, F., and Yu, D.: The impact of the terrain effect on land surface temperature variation based on Landsat-8 observations in mountainous areas, Int. J. Remote Sens., 40, 1808–1827, https://doi.org/10.1080/01431161.2018.1466082, 2019.

Jiménez-Muñoz, J. C., Sobrino, J. A., Skoković, D., Mattra, C., and Cristóbal, J.: Land Surface Temperature Retrieval Methods from Landsat-8 Thermal Infrared Sensor Data. IEEE Geosci. Remote S., 11, 1840–1843, https://doi.org/10.1109/LGRS.2014.2312032, 2014.

Ndossi, M. I., and Avdan, U.: Application of Open Source Coding Technologies in the Production of Land Surface Temperature (LST) Maps from Landsat: A PyQGIS Plugin, Remote Sens., 8, 413. https://doi.org/10.3390/rs8050413, 2016.

Wang, L., Lu, Y., and Yao, Y.: Comparison of Three Algorithms for the Retrieval of Land Surface Temperature from Landsat 8 Images, Sensors, 19, 5049, http://doi.org/10.3390/s19225049, 2019.

Yu, X., Guo, X., Wu, Z.: Land Surface Temperature Retrieval from Landsat 8 TIRS—Comparison between Radiative Transfer Equation-Based Method, Split Window Algorithm and Single Channel Method, Remote Sensing, 6, 9829-9852, https://doi.org/10.3390/rs6109829, 2014.

5) The results or visualization about the effect of canopy cover on microclimate variability should be improved (Line:191-195).

**Reply:** We agree with the reviewer that the part regarding the temperature variability should be strengthened. We have added a more thorough description of the results and edited figures of Tmean and Tmax to include standard deviation:

Tmean:

[Figure]

Tmax:

[Figure]

**Specific comments:**

Line 22-23: "......vary strongly with elevation and ambient temperatures". It's unclear to me, what're the ambient temperatures refer to? (if it is macroclimatic air temperature? I guess you did not measure this indicator)

**Reply:** Thank you for the comment, we agree that our description of macroclimate/ambient temperatures should be improved. With ambient temperatures, we mean the macroclimate, such as measured by weather stations. We did not use weather station data in our analysis to describe macroclimate, but rather reached a conclusion of the macroclimatic conditions from the temporal TOMST data (since macroclimate is a strong influencer on microclimate).

Line 24: what do the macroclimatic conditions mean?

**Reply:** Please see previous reply.

Line 78: how to quantify the stability of microclimates?

**Reply:** We have used standard deviation of mean temperatures during the study period to quantify the stability. We added a sentence in the methods to describe this approach and clarify what we have done.

Line 86-87: please provide the LAT, LON location information, although this could be interpreted from the Fig. 1

**Reply:** We added the LAT, LON information in the sentence.

Line 103-107: this information would be better to be assimilated into the introduction part

**Reply:** Thank you for the suggestion, we agree that the part fits better in the introduction and have moved it there.

Line 122-124: Please provide more information about microclimate sensors installed environments, they will affect collected microclimate data. For example, at low CC sites, the sensor was installed shadow area or sunlit area, and there are large temperature differences between sun and shade, meanwhile sunlit or shadow saturation also changes with sun angles, how did you consider these issues?

**Reply:** We acknowledge that microclimate is spatially variable in small scale, and that the placement of the sensor will affect the recorded temperatures. However, there is no perfect way to install the sensors, particularly when the number is limited. We put the sensors in places that represented the CC and environment as well as possible: in open areas they were exposed to the sun throughout the day, while in closed canopies they were shaded most of the day. In moderate CC, the sensors received both sunlight and shade, and we believe that the differences evened out during the day and would not detrimentally affect the results. We aimed to put sensors in places that were flat and not right next to trees or other artifacts. In some sites, the soil properties hindered us from putting the sensor in an optimal place, but we considered that the location we had to settle for was still adequate for the purpose. We have added a sentence about this in the methods section.

Line 129-131: macroclimate also largely affects microclimate (e.g. Fig. 1c in Zellweger et al., 2020), how do you consider this issue or how do you eliminate the influence of CC on microclimate? Meanwhile, whether the elevation and CC independent (independent assumption of the linear model)?

**Reply:** Please see reply to 1. We also chose field sites to have different canopy covers in different elevations to avoid multicollinearity of CC and elevation.

Line 134: please add the values.

**Reply:** Thank you for pointing out the mistake! We have added the missing values to the text.

Line 137-141: please clarify whether the mismatch of the collection date between ALS data and microclimate data could influence your research results, or evaluate its uncertainty?

**Reply:** Thank you for the comment. We accounted for the different data collection dates by comparing the ALS data to hemispherical photographs from the sites to see if the CC had changed substantially in 5-6 years. We could not find significant differences and therefore concluded that the data mismatch would not make a significant difference in the results. We have clarified this in the text and added a sentence and the following figure showing the differences in the Appendix A.

[Figure]

Line 159: how do you calculate the land surface emissivity?

**Reply:** We calculated emissivity based on an algorithm using the NDVI image (Ndossi & Avdan, 2016). Each pixel was given a pre-defined emissivity value based on the NDVI of the pixel:

NDVI < -0.185 → LSE 0.995; -0.185 ≤ NDVI < 0.157 → LSE 0.985; 0.157 ≤ NDVI ≤ 0.727 → LSE 1.009 + 0.047 * ln(NDVI); NDVI > 0.727 → LSE 0.990

We have added a description in the text.

Ndossi, M. I., and Avdan, U.: Application of Open Source Coding Technologies in the Production of Land Surface Temperature (LST) Maps from Landsat: A PyQGIS Plugin, Remote Sens., 8, 413. https://doi.org/10.3390/rs8050413, 2016.

Line 170-174: I am worried about this correction: this method could be regarded as three independent steps: the first step is removing elevation influence by minus dTh; the second step is removing slope influence by minus dTh; the third step is removing aspect influence by minus dTa. It makes sense if these three factors are independent, if they are highly correlated, this method will over-correct the LST?

**Reply:** Thank you for the excellent remark. We have removed equation 7 from the manuscript, because it described misleadingly what we did to correct for the topographic effect. There was not one method we used, but instead we estimated four different models, from where we have derived

our results. In the models, we used different approaches, and have accounted for the possible interactions of topographic variables and CC. Especially slope and elevation could be highly correlated; steep slopes are likely found most in the mountains. Moreover, aspect and slope can have high interaction, because the azimuth angle will affect LST. We tested the correlations between the topographic variables and CC, and none exceeded ±0.6. We have also reassessed our models and modified model 4 by including the interaction between slope and aspect classes instead of the interaction between CC and aspect. The model performance did not improve, and there was no big difference in the CC coefficient. The elevational effect of CC remained approximately the same.

Line 191: 'affected also' → 'also affected'; please give the full definition of SD when it first appears

**Reply:** Thank you for the remark. We changed the order of words and added the definition of SD.

Line 191-195: where are these results from? I guess maybe from the Figure, this information may not be obtained by readers directly. please clarify.

**Reply:** Please see reply to 5.

Line 258-261: what's the underlying reason the cooling impact of CC decease with elevation? Table 4 may be moved into supplementary

**Reply:** As temperatures decrease with lapse rate, also the demand for evapotranspiration and VPD decreases. The effect resembles the latitudinal shift from strong cooling in the tropics to strong warming in boreal regions. Consequently, on cooler days, plant evapotranspiration is lower than on hot days. The total incoming radiation is high on clear vs. cloudy days, and due to tree cover, the understory receives proportionally less radiation compared to open areas on sunny days. (Geiger, 1980; De Frenne et al., 2021). We have improved the manuscript by discussing the underlying mechanisms more.

We also moved Table 4 into the supplementary material.

References:

De Frenne, P., Lenoir, J., Luoto, M., Scheffers, B. R., Zellweger, F., Aalto, J., . . . Hylander, K.: Forest microclimates and climate change: Importance, drivers and future research agenda, Glob Chang Biol., 27, 2279–2297, https://doi.org/10.1111/gcb.15569, 2021

Geiger, R.: The climate near the ground, 4th edition, Harvard University Press, United States of America, 1980.

Line 286-287: Not clear to me, the sensible flux of canopy means the sensible heat exchange between canopy surface temperature and surrounding air temperature. Only when canopy temperature equals surrounding air temperature, we can say no sensible effect.

**Reply:** Thank you for the discernment. We agree on the definition of sensible heat. What we meant is a substantial or significant effect rather than sensible. We have corrected our poor choice of word and reformulated the sentence for more clarity.

**Reference:**

Zellweger, F., De Frenne, P., Lenoir, J., Vangansbeke, P., Verheyen, K., Bernhardt-Römermann, M., ... & Coomes, D. (2020). Forest microclimate dynamics drive plant responses to warming. Science, 368(6492), 772-775.

Zeng, Z., Wang, D., Yang, L., Wu, J., Ziegler, A. D., Liu, M., ... & Wood, E. F. (2021). Deforestation-induced warming over tropical mountain regions regulated by elevation. Nature Geoscience, 14(1), 23-29.

**Reply:** The suggested references were added to the manuscript.